# How IGF-1 activates its receptor

**Jennifer M Kavran[1], Jacqueline M McCabe[1], Patrick O Byrne[1], Mary Katherine Connacher[2], Zhihong Wang[2,3], Alexander Ramek[4], Sarvenaz Sarabipour[5], Yibing Shan[4], David E Shaw[4,6], Kalina Hristova[5], Philip A Cole[2], Daniel J Leahy[1,2]\***

[1]Department of Biophysics and Biophysical Chemistry, Johns Hopkins University School of Medicine, Baltimore, United States; [2]Department of Pharmacology and Molecular Sciences, Johns Hopkins University School of Medicine, Baltimore, United States; [3]Department of Chemistry and Biochemistry, University of the Sciences, Philadelphia, United States; [4]DE Shaw Research, New York, United States; [5]Department of Materials Science and Engineering, Johns Hopkins University, Baltimore, United States; [6]Department of Biochemistry and Molecular Biophysics, Columbia University, New York, United States

**Abstract** The type I insulin-like growth factor receptor (IGF1R) is involved in growth and survival of normal and neoplastic cells. A ligand-dependent conformational change is thought to regulate IGF1R activity, but the nature of this change is unclear. We point out an underappreciated dimer in the crystal structure of the related Insulin Receptor (IR) with Insulin bound that allows direct comparison with unliganded IR and suggests a mechanism by which ligand regulates IR/IGF1R activity. We test this mechanism in a series of biochemical and biophysical assays and find the IGF1R ectodomain maintains an autoinhibited state in which the TMs are held apart. Ligand binding releases this constraint, allowing TM association and unleashing an intrinsic propensity of the intracellular regions to autophosphorylate. Enzymatic studies of full-length and kinase-containing fragments show phosphorylated IGF1R is fully active independent of ligand and the extracellular-TM regions. The key step triggered by ligand binding is thus autophosphorylation.

**\*For correspondence:** dleahy@jhmi.edu

**Reviewing editor**: John Kuriyan, Howard Hughes Medical Institute, University of California, Berkeley, United States

## Introduction

The insulin and type-1 insulin-like growth factor receptors (IR and IGF1R) are homologous receptor tyrosine kinases (RTKs) that regulate cell metabolism, growth, and differentiation in a variety of mammalian tissues (*De Meyts, 2004*; *Siddle, 2011*, *2012*). Each is essential for normal development (*Liu et al., 1993*; *Accili et al., 1996*), and abnormal IR or IGF1R signaling is associated with many disorders, notably diabetes and cancer (*De Meyts and Whittaker, 2002*; *Pollak, 2012*). Although IGF1R is mainly associated with cell growth and differentiation and IR with regulation of glucose and lipid metabolism (*Siddle, 2012*), IGF1R and IR share 58% sequence identity and appear to signal via largely conserved molecular mechanisms (*Siddle, 2011*; *Ward et al., 2013*).

IR and IGF1R are disulfide-linked homodimers of single-pass integral membrane protein subunits. Each subunit undergoes a furin-like cleavage into α and β chains that remain disulfide-linked and are composed of six extracellular domains (L1, CR, L2, Fn1, Fn2, and Fn3) followed by a transmembrane (TM) region, a ~30 amino acid juxtamembrane region, a tyrosine kinase domain, and a C-terminal tail (*Ullrich et al., 1985*; *De Meyts and Whittaker, 2002*) (*Figure 1A,B*). An ~110 a.a. nonglobular insertion in Fn2, termed the insert domain (ID), contains the cleavage site and three cysteines of which one or more form reciprocal inter-subunit disulfide bonds. An additional inter-subunit disulfide is formed between cysteines in Fn1 (*Cheatham and Kahn, 1992*; *Schaffer and Ljungqvist, 1992*; *Sparrow et al., 1997*).

**eLife digest** Hormones are chemicals that are produced to carry signals around the body. Mammals, including humans, need hormones called insulin and insulin-like growth factors (or IGF for short) to grow and develop normally. These hormones bind to, and activate, specific proteins—known as receptors—that span from the outside of the cell to the inside through the cell's surface membrane.

The insulin and IGF receptors are complex molecules, each composed of two identical protein subunits that are linked together. The hormones bind to the part of the proteins (known as the extracellular domains) that are on the outside surface of cells. When no hormone is bound to the receptor, these two extracellular domains form an inverted 'V' shape and the two regions that cross the membrane are held far apart. It was unclear, however, how this shape changes when the hormone binds, and how this change in shape activates the receptor.

Kavran et al. have now compared the known three-dimensional structures of the extracellular domains of the insulin receptor, both with and without a molecule of insulin bound to it. This comparison highlighted an interaction between the two receptor subunits that was disrupted when insulin was bound to the receptor. This interaction appeared to stabilize the inverted 'V' structure and hold apart the parts of the proteins that span the surface—which in turn separates the regions of the proteins that are inside the cell. Kavran et al. suggest that this separation keeps the insulin receptor in an inactive state. Removing either the whole extracellular domain of the IGF receptor—or a smaller portion that keeps the regions that cross the cell membrane separate—resulted in the receptor being activated, even when insulin-like growth factor was not bound to the receptor.

Kavran et al. then confirmed that when a molecule of insulin-like growth factor binds to the extracellular domain of the IGF receptor, it causes a shape change that moves the parts of the receptor that span the cell membrane closer together. This results in the regions of the receptor proteins located inside the cell adding chemical tags, called phosphate groups, to one another—which activates the receptor.

When there are problems with how these receptors are activated or inactivated in humans, serious disorders including diabetes and cancer can occur. As such, the findings of Kavran et al. might help future work aimed at regulating the activation of the insulin and IGF receptors to treat these and other diseases.

Ligand binding to IR and IGF1R extracellular regions (ECDs) stimulates receptor kinase activity, leading to phosphorylation of multiple substrates and initiation of specific signaling cascades (*Siddle, 2012*). IR family members are unique among RTKs in forming constitutive dimers (of αβ subunits). Dimerization per se thus cannot be the activating signal, and activation is thought to involve a ligand-dependent conformational change (*Frattali et al., 1992*; *Lemmon and Schlessinger, 2010*). Part of the function of the ECD appears to be maintaining an inactive state in the absence of ligand as tryptic removal of the IR ECD results in constitutive activity (*Tamura et al., 1983*; *Shoelson et al., 1988*).

Two ligand-binding sites are present in each αβ dimer. Each site is composed of two distinct partial sites known as 'Site 1', which is composed of residues on L1 from one subunit and residues on the αCT' helix of the other subunit (a prime is used to indicate the opposite subunit), and 'Site 2', which is composed of residues on Fn1' and Fn2' (*Williams et al., 1995*; *Mynarcik et al., 1996*; *Whittaker et al., 2001*, *2008*; *Smith et al., 2010*) (*Figure 1—figure supplement 1*). A classic feature of ligand binding to IR and IGF1R is negative cooperativity (*De Meyts, 2004*), which implies communication between the two sites such that ligand binding to one site generates an asymmetric state of the receptor in which the affinity of the second site for ligand is weakened. The nature of this asymmetric state is not known, but hydrogen–deuterium exchange experiments with IGF1R identified regions that are likely foci of this asymmetry (*Houde and Demarest, 2011*).

Crystal structures of the unliganded ECD of IR (*McKern et al., 2006*; *Smith et al., 2010*) and a fragment of the IR ECD bound to Insulin (*Menting et al., 2013*) have greatly aided efforts to understand the molecular mechanisms governing IR/IGF1R activity. In the absence of ligand, the αβ subunits of the ECD form a symmetric, antiparallel dimer shaped like an inverted 'V' (*Figure 2A*). The apex

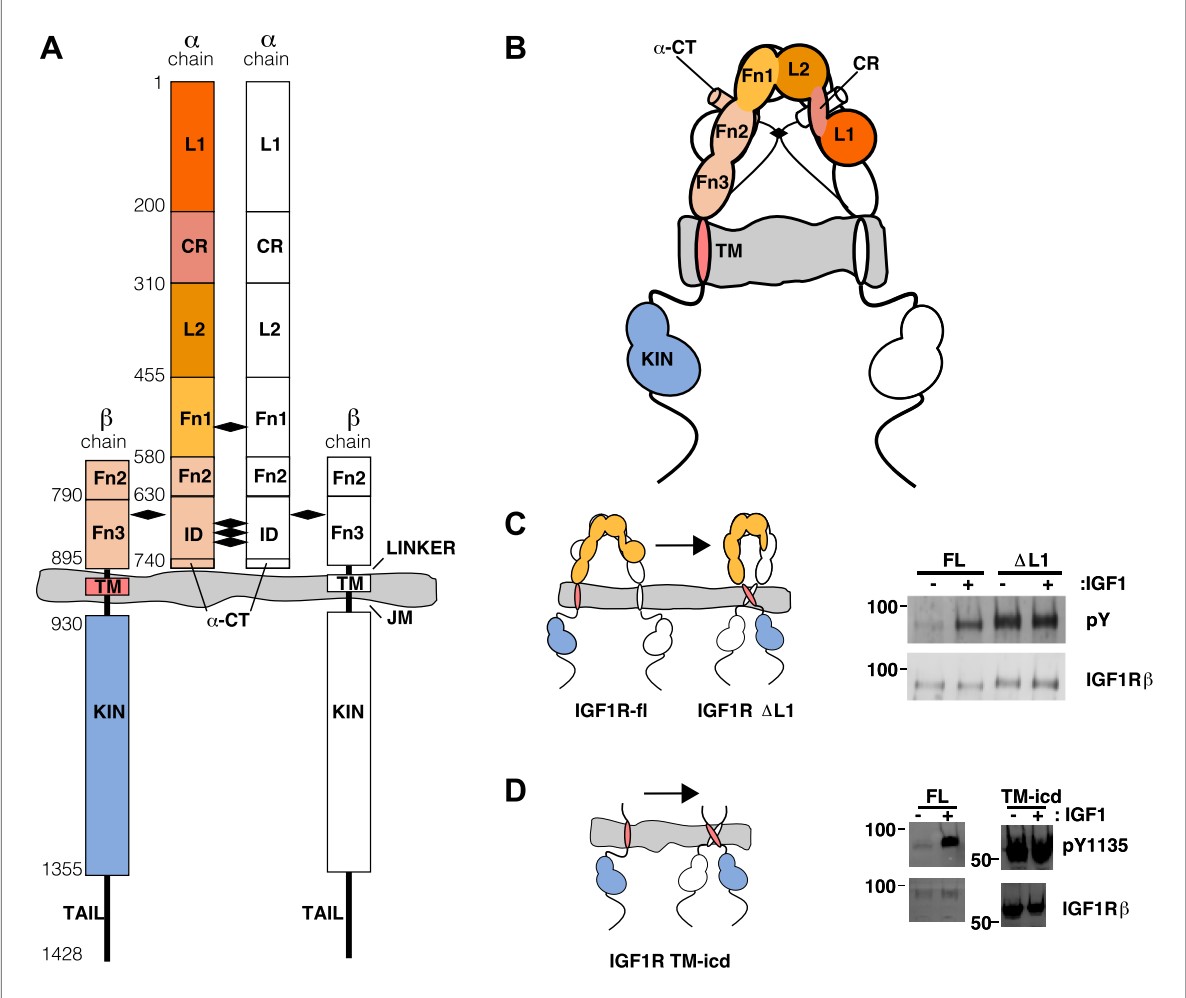

**Figure 1**. The extracellular domain of IGF1R autoinhibits IGF1R activity. (**A**, **B**) Schematic and cartoon representations of IGF1R. Numbering refers to human IGF1R excluding the signal sequence. Disulfide bonds are indicated by black diamonds. One αβ subunit is shown in outline and the other in color. (**C**) Cartoon of IGF1R full-length (IGF1R-fl) or an IGF1R fragment lacking the L1 domain (IGF1R ΔL1) (left). HEK293 cells expressing either IGF1R-fl (FL) or IGF1R ΔL1 (ΔL1) were incubated with or without IGF1. IGF1R proteins were immunoprecipitated from normalized cell lysates and levels of autophosphorylation or IGF1R expression detected by Western blot with anti-phosphotyrosine (pY) or anti-IGF1Rβ antibodies (right). Molecular weight standards indicated to the left of each panel (**D**). Diagram of IGF1R variant lacking the ECD (IGF1R TM-icd) (left). Western blots probed with anti-pY1135, which recognizes a phosphotyrosine on the activation loop of the IGF1R kinase domain, and anti-IGF1Rβ of normalized lysates from cells expressing IGF1R-fl (FL) or IGF1R TM-icd (TM-icd), in the presence and absence of IGF1 (right). All panels are from the same blot. See also supplemental figures.

The following figure supplements are available for figure 1:

**Figure supplement 1**. Residues on IR important for ligand binding.

**Figure supplement 2**. Phosphorylation of the ICD.

**Figure supplement 3**. Activity assay of IGF1R TM-icd.

of this 'V' is formed by reciprocal interactions between L2-Fn1 domain pairs from opposing subunits (*Figure 2C*), with the Fn2 and Fn3 domains (Fn2–3) forming 'legs' with C-termini separated by 117 Å. The fragment of IR crystallized with Insulin encompassed the first four domains of the ECD (L1-CR-L2-Fn1) with the αCT peptide fused to the C-terminus of Fn1. The structure of this fragment revealed the interaction between Site 1 and Insulin and rationalized a wealth of biochemical data. The nature of the activating conformational change in IR remained unclear, however, owing to the absence of the

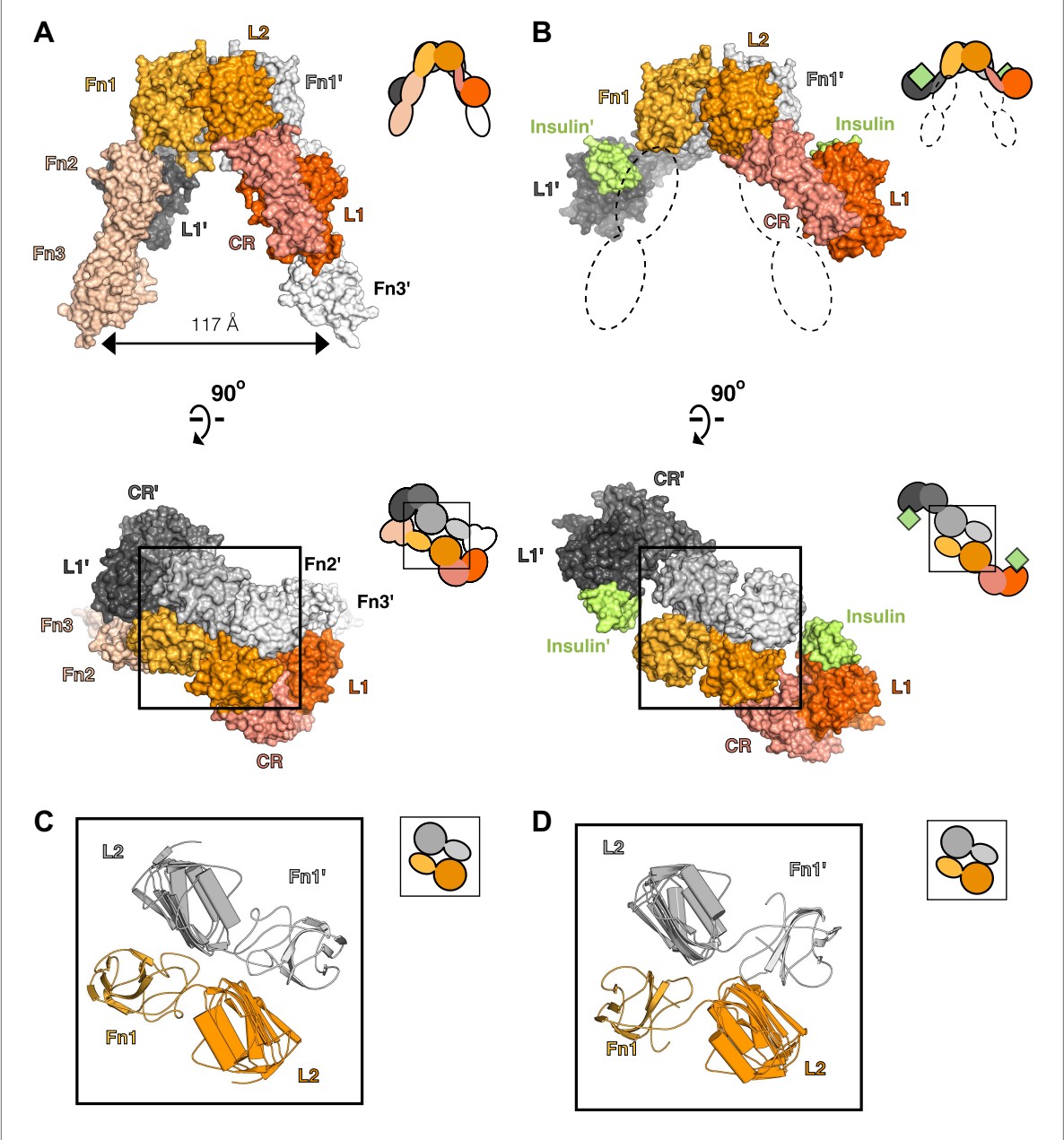

**Figure 2**. Conserved IR ECD dimer interaction. Surface representations of (**A**) the unliganded IR ECD dimer (PDB 3LOH; *Smith et al., 2010*) and (**B**) the IR ECD fragment bound to Insulin (PDB 3W14; *Menting et al., 2013*). Domains of one αβ subunit are colored in shades of orange, the other in shades of gray, and Insulin in light green. N-terminal domains are in darker shades. The conserved L2-Fn1:L2'-Fn1' interface is marked by a black box. Ribbon diagrams of these domain pairs are shown for the unliganded (**C**) and liganded (**D**) IR ECD dimer. A cartoon of domains shown are included at the upper right of each structure. Dashed lines represent the missing Fn2-3 domain tandems.

Fn2-ID-Fn3 regions, which contain Site 2, and the presence of an apparently nonphysiological dimer mediated by αCT regions (*Menting et al., 2013*). Current models of IR/IGF1R activation envision only minor structural rearrangements of the ECD when ligand binds and posit a ligand-dependent twist of the Fn2–3 domains with little change in TM separation as the activating signal (*Ward and Lawrence, 2012*; *Ward et al., 2013*).

Although structural and mechanistic details of ligand-dependent activation of the receptor remain unclear, autophosphorylation of the kinase domain is a key element of receptor activation. Early studies

with purified IR showed that autophosphorylation enhances IR kinase activity and that addition of Insulin led to a threefold to fivefold increase in $V_{max}$ with no change in $K_m$ for peptide substrate (**Kasuga et al., 1983**; **Pike et al., 1986**). More recent studies with isolated IGF1R kinase domains showed little phosphorylation of peptide substrates in the absence of autophosphorylation but increased activity with each successive trans-autophosphorylation of the three tyrosines in the kinase activation loop (**Favelyukis et al., 2001**) (**Figure 1—figure supplement 2**). Crystal structures of IR and IGF1R kinase domains in both inactive and active, phosphorylated conformations revealed canonical kinase domain features and rationalized how autophosphorylation promotes the active conformation (**Hubbard et al., 1994**; **Hubbard, 1997**; **Favelyukis et al., 2001**; **Huse and Kuriyan, 2002**; **Munshi et al., 2002**). A specific asymmetric dimer of kinase regions is necessary for activation of the Epidermal Growth Factor Receptor (EGFR) (**Zhang et al., 2006**), a related RTK, but whether or how kinase domain interactions play a role in regulating IR/IGF1R activity is not known.

To address the gaps in our understanding of the molecular mechanisms underlying IR family activation, we inspected crystal structures of the IR ECD for clues to the nature of conformational changes that occur when ligand binds. We identified an underappreciated dimer of the IR ECD fragment with Insulin bound that preserves key inter-subunit interactions present in the unliganded IR ECD and illuminates conformational changes that occur when ligand binds. In particular, an inter-subunit interaction that stabilizes the separation of the Fn2–3 'legs' of the ECD is disrupted by ligand binding, suggesting that TM separation may be important for maintaining the receptor in an inactive state. We tested this model in a series of biochemical and biophysical assays and show that the IGF1R ECD indeed autoinhibits activity by holding the TMs apart in the absence of ligand. Ligand binding releases this autoinhibition and allows the TMs to come together in an autophosphorylation-competent state. Enzymatic activity assays of purified, full-length IGF1R and several kinase-domain containing IGF1R fragments further show that, once phosphorylated, IGF1R is fully active independent of ligand or allosteric stimulation. The key step regulated by ligand binding is thus autophosphorylation and not kinase activity per se, and the role of the IR/IGF1R ECD is to inhibit activity in the absence of ligand rather than promote activity in the presence of ligand.

## Results

### Ligand binding disrupts interactions stabilizing IGF1R/IR TM separation

To search for ligand-dependent conformational changes in the IR ECD, we compared the crystal structures of liganded and unliganded IR ECD fragments (**Smith et al., 2010**; **Menting et al., 2013**). The reported dimer of the liganded IR fragment is mediated only by reciprocal interactions between L1 from one subunit and αCT from the other. As this dimer neither resembles the inverted 'V' conformation of the unliganded IR ECD nor is compatible with the inter-subunit disulfide bond between Fn1 domains, it is likely a nonphysiological artifact arising from the deletion of Fn2-ID-Fn3 domains and fusion of αCT to Fn1 (**Menting et al., 2013**). We therefore examined all IR interactions present in the crystal and identified an alternative dimer (**Figure 2B**), which had previously been noted as a part of a dimer of dimers (**Menting et al., 2013**), that is compatible with the disulfide bond between Fn1 domains (**Schaffer and Ljungqvist, 1992**) and preserves the inter-subunit contacts between opposing L2-Fn1 domains (L2-Fn1:L2'-Fn1') present at the apex of the inverted 'V' in the unliganded IR structure (**Figure 2**). Superposition L2-Fn1:L2'-Fn1' domains from the unliganded IR structure (**Smith et al., 2010**) and this symmetry-generated dimer of liganded IR (**Menting et al., 2013**) results in a root mean square deviation of 3.4 Å for 478 Cα atoms (**Figure 2C,D**).

Using this superposition as a basis for comparing liganded and unliganded IR conformations reveals that the L1-CR domains rotate about a hinge between CR and L2 (**Figure 3A,B**) when ligand binds. The position of this hinge is homologous to that of the hinge observed in the ECD of EGFR when ligand binds, and most of the ~50° rotation can be ascribed to a rotation about the psi angle of Gly 306 (**Burgess et al., 2003**; **Menting et al., 2013**). The rigid-body rotation of the L1-CR domains repositions them further from the central axis of the IR dimer, exposes several residues on L1 that mediate Insulin binding, and disrupts an extensive interface between L1 and Fn2'-3' domains in unliganded IR (**Figure 3C,D**) (**Hubbard, 2013**; **Menting et al., 2013**). This interface fixes the relative positions of the Fn2–3 tandems in unliganded IR and stabilizes the ~120 Å separation of the Fn3 domains (and the immediately adjacent TM regions) observed in the absence of ligand.

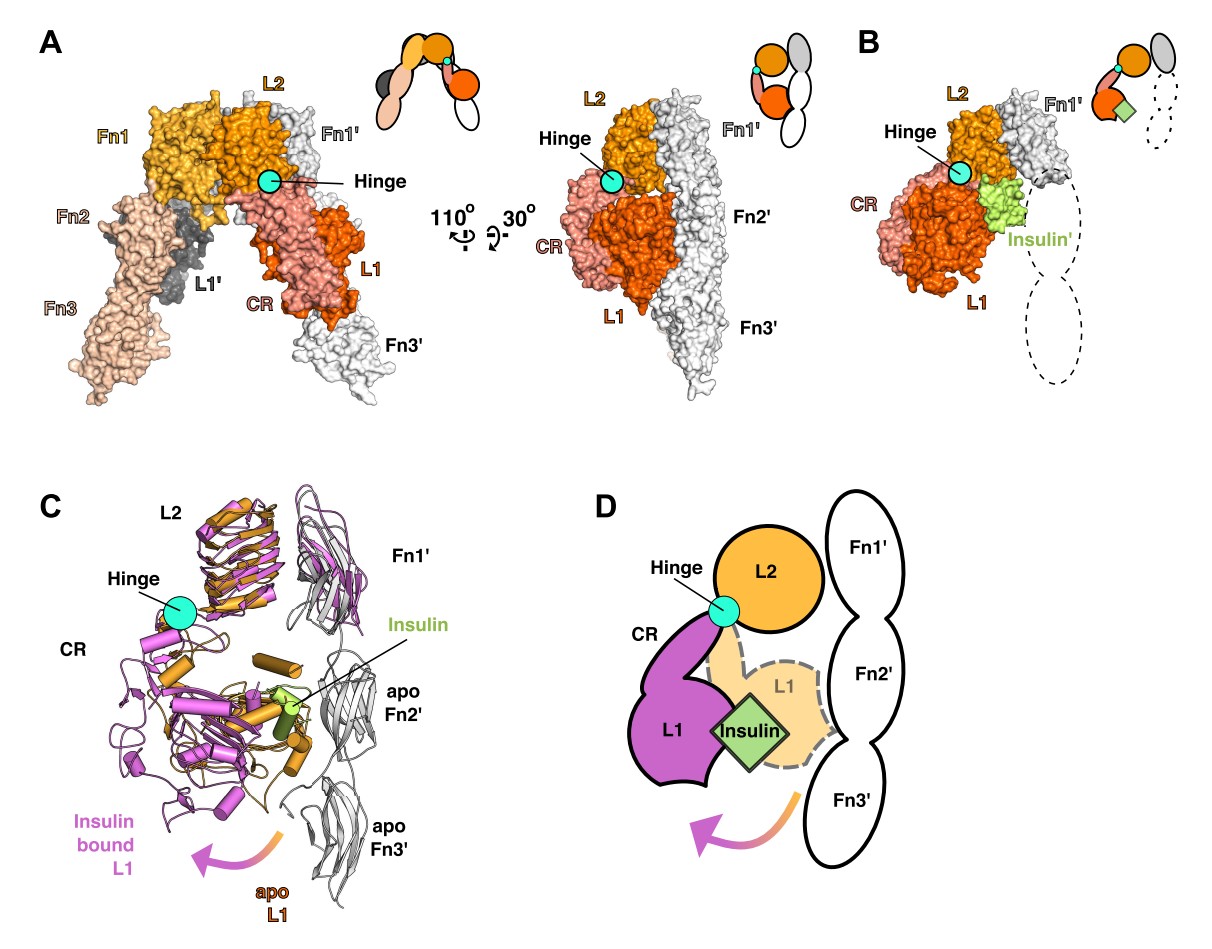

**Figure 3**. Ligand-induced conformational change in the IR ECD. (**A**) Surface representation of the unliganded IR ECD (PDB 3LOH; *Smith et al., 2010*) oriented to show the L1:Fn2'-3' interface. The hinge between CR and L2 is indicated by a cyan circle. Cartoons of domains shown are to the upper right of each structure. (**B**) Surface representation of ligand-bound IR ECD fragment (PDB 3W14; *Menting et al., 2013*) in the same orientation as in **A**. The missing Fn2-3 domains are indicated by dashed lines. (**C**) Superposition of the ligand-bound IR structure (purple ribbon) onto the unliganded structure (orange and white ribbon) using the L2-Fn1 domain pairs. Insulin is displayed in light green. An arrow shows the direction of the relative movement of unsuperposed L1-CR domains about a hinge (cyan circle) between the CR and L2 domains. Complementary halves of each receptor subunit were omitted for clarity. (**D**) A cartoon showing the movement of the L1-CR domain tandems.

This observation led us to ask whether the TM separation enforced by the L1:Fn2'–3' interface maintains the receptor in an inactive state in the absence of ligand. To test this possibility, we substituted several L1 residues that are both conserved between IGF1R and IR and occur at the L1:Fn2'–3' interface with residues of opposite charge or increased size. These substitutions consistently resulted in poor processing and/or expression of IGF1R, however, which precluded analysis of their effects on IGF1R activation. We then analyzed an IGF1R variant lacking the entire L1 domain (IGF1R ΔL1) and found that it expressed well in HEK293 cells and was constitutively phosphorylated as judged by anti-phosphotyrosine Western blots (*Figure 1C*), demonstrating that the L1 domain is essential for maintaining the inactive state of IGF1R.

## The IGF1R extracellular region is autoinhibitory

As deletion of the L1 domain results in constitutive activity of IGF1R, we next tested whether the role of the ECD might be to stabilize an inactive state in the absence of ligand as opposed to stabilizing the active state in the presence of ligand. We deleted most or all of the IGF1R ECD and found that these truncated IGF1Rs (IGF1R Fn3-TM-icd and IGF1R TM-icd) are constitutively phosphorylated in the absence of ligand (*Figure 1D* and *Figure 1—figure supplement 3*). A

kinase-inactivating point mutation eliminated this phosphorylation, implicating autophosphorylation as the source of this signal (*Figure 1—figure supplement 3*). The IGF1R ECD thus plays an active role in keeping the receptor off in the absence of ligand but is not needed to activate the truncated receptor.

## Release of IGF1R autoinhibition by ligand brings the TMs together

Loss of either the L1 domain or the ECD eliminates the inter-subunit interaction between L1:Fn2′–3′ that fixes the separation of the TM regions. To examine if TM separation is altered in active receptors, we replaced the intracellular domain (ICD) of IGF1R with fluorescent donor or acceptor proteins (IGF1R ECD-TM-fp) and determined the FRET efficiency between co-transfected donor–acceptor pairs in the presence and absence of ligand (*Figure 4A*). We used spectrally resolved Förster resonance energy transfer (FRET) (*Raicu et al., 2008*) to calculate the FRET efficiency at the pixel level in transfected CHO cells, which ensures only proteins expressed at the cell membrane are included in the analysis. FRET efficiency increased twofold in the presence of ligand (*Figure 4A*). Assuming comparable if not random orientations of the fluorescent proteins in both states, these data indicate the C-termini of the TMs move closer together in the presence of ligand. A similar increase in FRET efficiency was observed in vesicles derived from CHO cells expressing IGF1R ECD-TM-fp in the presence and absence of ligand (*Figure 4—figure supplement 1–3*). We also measured the FRET efficiency for IGF1R lacking the L1 domain (IGF1R ECDΔL1-TM-fp) and found that its FRET efficiency is similar to that of IGF1R ECD-TM-fp with bound ligand suggesting that the ligand bound state of IGF1R is similar to that of the receptor when the L1:Fn2′–3′ interface is disrupted (*Figure 4—figure supplement 4*).

If the IGF1R ECD inhibits IGF1R activity by enforcing TM separation, introduction of flexible linkers between the ECD and TM regions should release this inhibition, and insertion of progressively longer linkers composed of glycine–serine containing repeats between 4 and 20 residues long indeed led to increasing levels of basal receptor activity (*Figure 4B*).

## Isolated IGF1R TMs dimerize

The ligand-dependent decrease in distance between IGF1R TMs observed in FRET assays of IGF1R ECD-TM-fp led to the question of whether IGF1R TMs physically associate in the active state. Analysis of IR family TM sequences failed to identify any motifs known to mediate TM interactions, such as GXXXG (*Russ and Engelman, 2000*), but did identify an absolutely conserved proline (P911 in human IGF1R) (*Figure 5A* and *Figure 5—figure supplement 1*). To investigate the role of P911 in receptor activation, we performed cell based activity assays with IGF1R-fl bearing either a single P911L substitution or with additional substitutions to residues surrounding P911. These IGF1R variants became phosphorylated in a ligand-dependent manner suggesting P911 does not play a key role in mediating receptor activity, an outcome consistent with the results of earlier studies involving more dramatic manipulations of IR TM residues and suggesting lax constraints on TM sequence (*Frattali et al., 1991*; *Yamada et al., 1992*; *Cheatham et al., 1993*) (*Figure 5—figure supplement 2*).

To investigate the behavior of IGF1R TMs further, we performed molecular dynamics (MD) simulations of IGF1R TM monomers inserted into a 15% phosphatidylserine, 85% phosphatidylcholine bilayer. Four simulations extending between 12 μs and 36 μs were performed. In two of these simulations IGF1R TMs formed dimers in which the two TM helices cross each other opposite the conserved proline (*Figure 5B* and *Figure 5—figure supplement 3*). These dimers formed early in the simulations and were stable throughout the rest of the runs. The IGF1R TMs also adopted several stable monomeric conformations in these simulations, occasionally with a sharp kink at P911 (*Figure 5B*).

Experimental evidence that IGF1R TMs have an intrinsic propensity to dimerize in bilayers was obtained from quantitative imaging FRET (QI-FRET) experiments (*Li et al., 2008*). Isolated IGF1R TMs were fused to C-terminal donor or acceptor fluorescent proteins (IGF1R TM-fp) and transiently transfected into CHO cells (*Figure 4—figure supplement 5*). Vesicles containing variable amounts of expressed proteins were generated and FRET efficiency analyzed (*Chen et al., 2010a*; *Del Piccolo et al., 2012*) (*Figure 4—figure supplement 1–3*). FRET efficiency increased as a function of fluorescent protein concentration and exceeded both the theoretical levels of FRET expected to arise from random proximity (*King et al., 2014*) and the FRET efficiency of a control monomeric membrane protein (ErbB2 ECD-TM-fp) demonstrating an intrinsic propensity of IGF1R TMs to associate (*Figure 5C*). When examined on the surface of cells the FRET efficiency of IGF1R TM-fp overlapped with the FRET

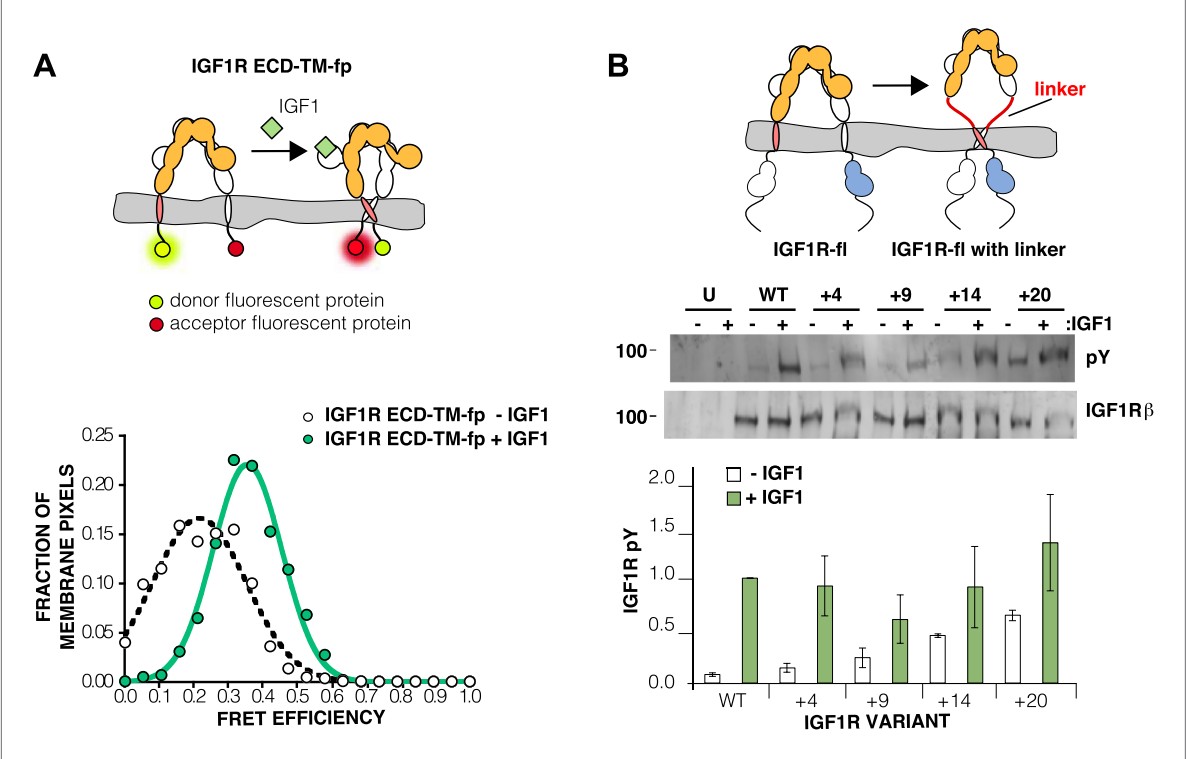

**Figure 4**. The IGF1R ECD maintains TM separation. (**A**) Cartoon of the IGF1R ECD-TM-fp variant utilized in the live cell FRET assay (top). Frequency distribution of FRET efficiency values per membrane pixel of IGF1R ECD-TM-fp in the presence (green) or absence (white) of IGF1 (bottom). Data were fit to a normal distribution with peak values of 0.34 ± 0.09 and 0.21 ± 0.13 in the presence and absence of IGF1, respectively. (**B**) Cartoon of the IGF1R-fl and IGF1R variants with flexible linkers (top). Representative western blots probed with either anti-pY and anti-IGF1Rβ antibodies of immunopreciptated, normalized cell lysates of untransfected cells (U) or cells transfected with IGF1R-fl (WT) or IGF1R variants with an additional 4 (+4), 9 (+9), 14 (+14), or 20 (+20) glycine and serine residues per αβ chain (middle). Bar graph of average IGF1R phosphorylation normalized to total receptor concentration (±s.e.m.) from at least three separate experiments, except for +14 which was from 2. Results from cells incubated in the absence of IGF1 are shown in white and in the presence of IGF1 shown in green (bottom). See also supplemental figures.

The following figure supplements are available for figure 4:

**Figure supplement 1**. Apparent FRET efficiency measured in membrane-derived vesicles.

**Figure supplement 2**. Ratio of donor to acceptor molecules in membrane-derived vesicles.

**Figure supplement 3**. Intrinsic FRET efficiency in vesicles, corrected for proximity FRET and varying donor-to-acceptor ratios.

**Figure supplement 4**. The TMs associate in IGF1R ECD-ΔL1-TM-fp.

**Figure supplement 5**. FRET efficiency in live cells of isolated TMs.

efficiency observed for IGF1R ECD-TM-fp in the presence of ligand (*Figure 4—figure supplement 5*) suggesting similar TM separations in liganded IGF1R and isolated IGF1R TM dimers.

An intrinsic propensity of IGF1R TM and ICD regions to dimerize was also demonstrated by cysteine substitutions in constitutively active ECD-truncated forms of IGF1R (IGF1R TM-icd) (*Figure 1—figure supplement 3*, *Figure 5D*). The side chains of the extracellular membrane-proximal residue H905 approach within 6 Å of one another in the MD dimer (*Figure 5B*), and H905 or T898 were individually substituted with cysteine in IGF1R TM-icd variants. Western blots of reduced and non-reduced lysates from HEK293 cells expressing these proteins show that the majority of the H905C variant forms a disulfide-linked dimer (*Figure 5D*). Lower amounts of dimerization were observed for the T898 variant in which the cysteines are farther away from the TM region (*Figure 5D*), and no disulfide

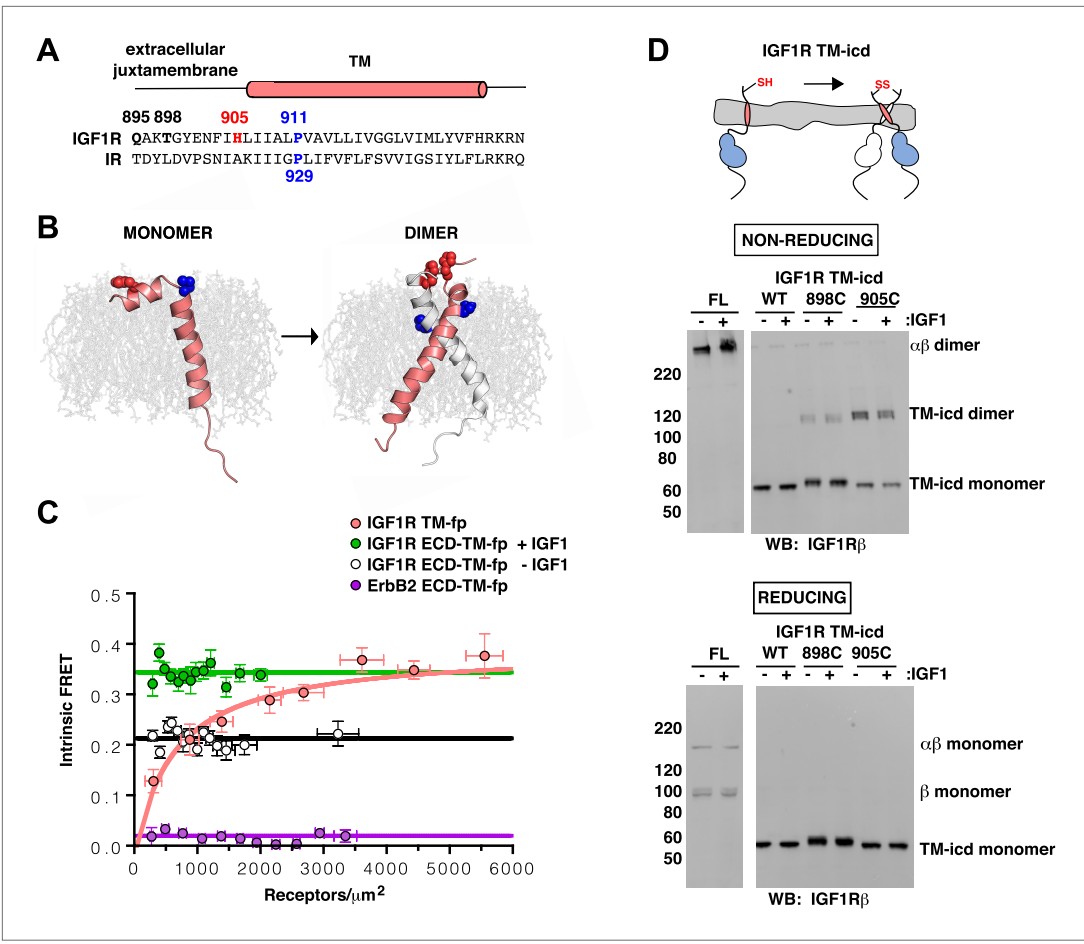

**Figure 5**. IGF1R TMs associate. (**A**) Sequence alignment of human IGF1R and IR extracellular juxtamembrane and TM regions. The position of the TM is indicated above the alignment. Bold lettering highlights residues targeted for substitution. H905 (red) and P911 (blue) are colored. (**B**) IGF1R TM monomer and dimer structures observed in MD simulations are shown in ribbon (white and pink). Shown in spheres are P911 (blue) and H905 (red). The lipid bilayer is in light gray. (**C**) The intrinsic FRET efficiency for IGF1R TM-fp plotted (pink) as a function of TM concentration. Also shown for reference are a monomeric control (ErbB2 ECD-TM-fp) in purple and the IGF1R ECD-TM-fp in the presence (green) or absence (white) of ligand. Each data point represents the binned average of at least eight vesicles (±s.e.m. in both x and y). For IGF1R TM-fp, the data were fit to a two state association model with a peak value of 0.38 ± 0.02. (**D**) Cartoon of IGF1R TM-icd with the positions of single-site cysteine mutations indicated in red (top). Anti-IGF1Rβ Western blots of lysates of HEK293 cells expressing either IGF1R-fl or IGF1R TM-icd fragments (WT, 898C or 905C) incubated with or without IGF1 and analyzed on non-reducing (middle) or reducing gels (bottom).

The following figure supplements are available for figure 5:

**Figure supplement 1**. Conservation of extracellular juxtamembrane and TM regions in IR family members.

**Figure supplement 2**. P911 is not required for proper IGF1R activation.

**Figure supplement 3**. Stable IGF1R TM dimers form during MD simulations.

crosslinking was observed for an equivalently truncated IGF1R without a cysteine substitution. These results demonstrate that, in the absence of the ECD, the IGF1R TM-icds come together in a manner consistent with the MD TM dimer.

## IGF1R TMs dimerize in the active, full-length receptor

We next investigated whether juxtamembrane cysteines formed disulfide crosslinks when introduced into IGF1R-fl (**Figure 6A**). We could not directly analyze intra-dimer disulfide bond formation by

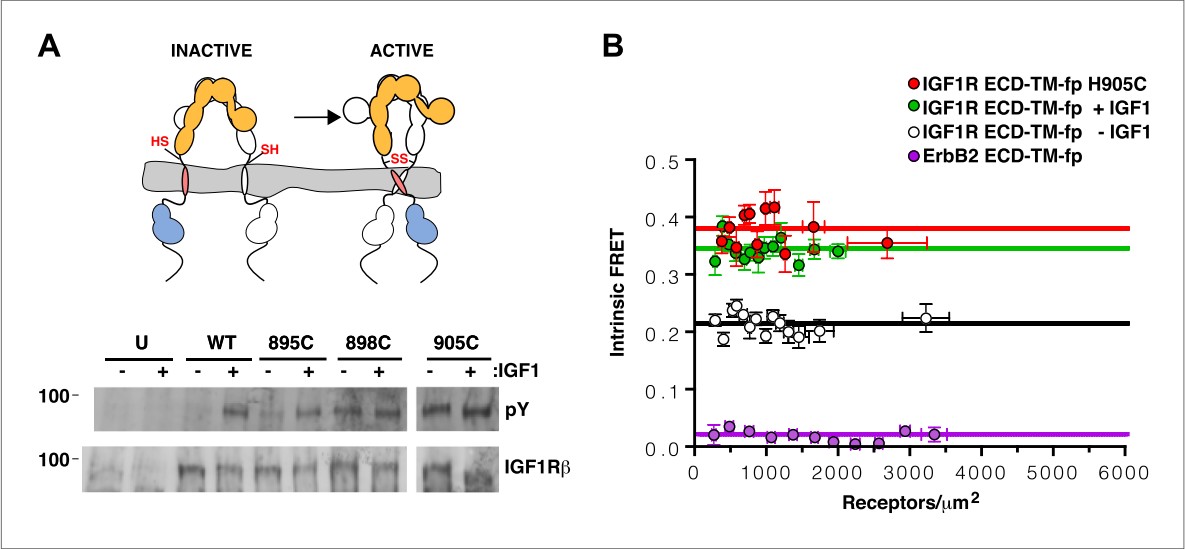

**Figure 6**. TMs associate in active IGF1R-fl. (**A**) Cartoon of IGF1R-fl with cysteine substitutions (top). Western blots of immunoprecipitated normalized cell-lysates from untransfected cells (U) or cells transfected with IGF1R-fl (WT) or IGF1R-fl proteins with cysteine substitutions (895C, 898C, or 905C) incubated the presence or absence of IGF1 (bottom). All panels are from the same blot. (**B**) Plot of the intrinsic FRET efficiency of IGF1R ECD-TM-fp with the H905C substitution (red) and fit to a horizontal line with a value of 0.38 ± 0.02. Each data point represents the binned average of at least eight vesicles (±s.e.m. in both x and y). For reference, the intrinsic FRET efficiencies and fits for IGF1R ECD-TM-fp with ligand (green) or without ligand (white) and the ErbB2 ECD-TM-fp negative control (purple) are also shown. See also supplemental figures.

The following figure supplement is available for figure 6:

**Figure supplement 1**. Matching migration patterns of wild-type and H905C IGF1R-fl.

introduced cysteines by analyzing migration on reducing and non-reducing SDS-PAGE owing to the native disulfide bonds linking α and β chains. To circumvent this issue, we substituted the cysteine residues that mediate the αβ disulfide with serine. These substitutions inhibited proper proteolytic processing of the receptor, however, obscuring differentiation of the new disulfide bond from native disulfide bonds. To our surprise, however, Western blots monitoring the phosphorylation of the IGF1R variants with the single substitutions of either T898C or H905C revealed high levels of phosphorylation in the absence of ligand (*Figure 6A*). An IGF1R-fl variant with a cysteine substitution immediately following the Fn3 domain and further away from the TM (Q895C) displayed only a modest increase in basal phosphorylation compared to wild-type (*Figure 6A*), perhaps owing to steric interference from Fn3 domains.

The most likely explanation for the ligand-independent activity of IGF1R variants with single cysteine substitutions is that IGF1R samples the active conformation in the absence of ligand and becomes trapped in the active state by a disulfide bond. An intra-dimer disulfide bond could not occur at these positions if the receptor was locked in the inactive 'legs' apart state by the L1:Fn2'–3' interaction. This interaction must open transiently for ligand to bind, however, and it appears this transient opening allows TM association. Alternatively, inter-dimer disulfides could form and activate the receptor. To distinguish these possibilities, we compared the migration of wild-type and IGF1R H905C on reducing and non-reducing SDS-PAGE. If the introduced cysteines form inter-dimer disulfide bonds, some fraction of IGF1R H905C would migrate as a dimer of dimers or higher (molecular weight ≥ ~640 kDa vs ~320 kDa for a dimer) on non-reducing SDS-PAGE, and the ratio of H905C dimer to total monomer would decrease relative to wild-type IGF1R. Western blots show that this ratio remains the same for H905C and wild-type IGF1R, however, indicating the absence of appreciable inter-dimer disulfide formation by IGF1R H905C (*Figure 6—figure supplement 1*). Sequence alignments show that H905, and the IGF1R juxtamembrane region in general, can be substituted with at least eleven different amino acids indicating the effects of cysteine at this site likely stem from its ability to form disulfide bonds (*Figure 5—figure supplement 1*).

An intra-dimer disulfide between cysteines at position 905 would force TM association. When the H905C substitution was introduced into IGF1R ECD-TM-fp, the FRET efficiency of this variant in the

absence of ligand matched that of native IGF1R ECD-TM-fp in the presence of ligand (*Figure 6B*). This result shows that the H905C substitution indeed leads to decreased TM separation, presumably owing to crosslinked juxtamembrane regions, and that this conformation is indistinguishable from that observed for liganded IGF1R ECD-TM-fp.

## Ligand binding does not stimulate the kinase activity of phosphorylated IGF1R

To understand how changes in IGF1R ECD conformation are linked to receptor kinase activity, we next assessed the effects of ligand binding and autophosphorylation on the kinase activity of IGF1R. We expressed full-length IGF1R (IGF1R-fl) in HEK293 cells and purified it to near homogeneity for in vitro kinase assays (*Figure 7—figure supplement 1,2*). To parse the contributions of specific regions to regulation of IGF1R activity, we expressed the isolated kinase domain (IGF1R-kin), the N-terminal jux-tamembrane segment linked to the kinase domain (IGF1R-jmk), and the entire intracellular domain (IGF1R-icd) in Sf9 cells and purified each fragment. Either fully phosphorylated or unphosphorylated (or minimally phosphorylated) versions of each form of IGF1R were prepared by incubation with ATP and Mg$^{++}$ or phosphatase, respectively, during purification (*Figure 7—figure supplement 3*).

The steady-state kinetic parameters of each form of IGF1R were determined using a direct, radio-metric assay monitoring phosphotransfer from [γ-$^{32}$P]ATP to a biotinylated peptide substrate (*Table 1*) (*Table 1—source data 1*). Activity in this assay was linear with respect to both time and enzyme concentration for all forms of IGF1R (*Table 1—source data 2, 3*). Each intracellular IGF1R fragment showed barely detectable catalytic activity when unphosphorylated, which differed from the low but measurable activity of phosphatase-treated IGF1R-fl (*Table 1*). Western blots indicated incomplete dephosphoryla-tion of IGF1R-fl despite efforts to drive this reaction to completion, however, and residual autophospho-rylation likely accounts for the elevated activity of IGF1R-fl relative to unphosphorylated IGF1R fragments.

In contrast, robust activity was observed for all phosphorylated forms of IGF1R. $K_m^{app}$ values for both ATP and peptide ranged between 50 and 110 µM, and $k_{cat}$ values ranged between 98 and 749 min$^{-1}$

**Table 1.** Enzymatic Parameters of different length IGF1R proteins‡§

|  | enzyme pY | $K_m^{app}$ ATP (µM) | $K_m^{app}$ Peptide (µM) | $k_{cat}$ (min$^{-1}$) | $k_{cat}/Km^{app}{}_{(ATP)}$ |
|---|---|---|---|---|---|
| IGF1R-fl + IGF1 | + | 82 ± 11 | 48 ± 8 | 459 ± 17 | 5.6 |
| IGF1R-fl + IGF1 | − | 356 ± 61 | 68 ± 19 | 44 ± 3 | 0.1 |
| IGF1R-fl | + | 89 ± 18 | 91 ± 8 | 749 ± 22 | 8.5 |
| IGF1R-fl | − | 172 ± 35 | 66 ± 17 | 62 ± 5 | 0.4 |
| IGF1R-icd | + | 79 ± 9 | 110 ± 12 | 385 ± 13 | 4.9 |
| IGF1R-icd | − | >1000* | >1000* | N.D.† | 0.0013 |
| IGF1R-jmk | + | 68 ± 12 | 52 ± 5 | 363 ± 20 | 6.9 |
| IGF1R-jmk | − | >1000* | >1000* | N.D. † | 0.0029 |
| IGF1R-kin | + | 115 ± 24 | 114 ± 26 | 98 ± 6 | 0.85 |
| IGF1R-kin | − | >2000* | >1250* | N.D. † | 0.0019 |

*The absolute value of *Km* could not be determined but is greater than value stated.
†$k_{cat}$ value could not be measured.
‡Enzymatic values (±s.d.) calculated from duplicate experiments are shown.
§See also supplemental figures.

**Source data 1**. Representative curves of steady-state kinetic analyses for each IGF1R protein characterized. Each data point was performed in duplicate and is shown separately.

**Source data 2**. Enzyme behavior is linear with respect to time. Product/Enzyme plotted vs time (minutes) for each IGF1R protein investigated.

**Source data 3**. Enzyme behavior is linear with respect to enzyme concentration. Velocity (nM of product/min) plotted vs enzyme concentration (nM) for each IGF1R protein investigated.

(*Table 1*). Addition of the juxtamembrane region (IGF1R-jmk) resulted in fourfold increase in $k_{cat}$ from that observed for IGF1R-kin, but addition of the C-tail (IGF1R-icd) had no effect on activity (*Table 1*) (*Figure 7A*). Notably, $k_{cat}$ differences of less than twofold were observed between phosphorylated forms of IGF1R-fl and either IGF1R-jmk or IGF1R-icd indicating that the activities of these phospho-proteins are essentially equivalent. That is, the C-tail, ECD, and TMs do not contribute to the activity of phosphorylated IGF1R. Surprisingly, the $k_{cat}$ of phosphorylated IGF1R-fl is marginally less with ligand than without indicating that bound ligand also does not contribute to the activity of phosphorylated IGF1R-fl.

## Ligand binding stimulates IGF1R autophosphorylation

Since the activity of phosphorylated IGF1R-fl is high and unaffected by ligand binding, we reasoned that autophosphorylation must be the step controlled by ligand-dependent allostery. To test this idea, phosphatase-treated IGF1R-fl was incubated with ATP and Mg++ in the presence and absence of IGF1, and the rate of autophosphorylation monitored by Western blotting. The presence of IGF1 indeed led to a marked increase in the rate of autophosphorylation (*Figure 7B*).

## No evidence for specific intracellular domain interactions

Interactions between kinase domains have proven essential for stabilizing the active state of EGFR (*Zhang et al., 2006*), and we wondered if specific interactions between kinase domains are coupled to active or inactive states in the IR family. To address this question, we analyzed lattice contacts in crystals of active and inactive IGF1R and IR kinases to determine if any contacts appear repeatedly or bear hallmarks of a physiological interaction. Several of nearly 300 unique contacts identified bury more than 800 Å² of surface area (*Bahadur et al., 2004*) but none appeared in multiple crystal forms or suggest a rationale for maintaining a stable active or inactive state (*Figure 8—figure supplement 1*).

We next used mutagenesis and a cell-based activity assay to search for surfaces on the IGF1R kinase important for maintaining an inactive or active state. Six clusters of 3–9 alanine substitutions that

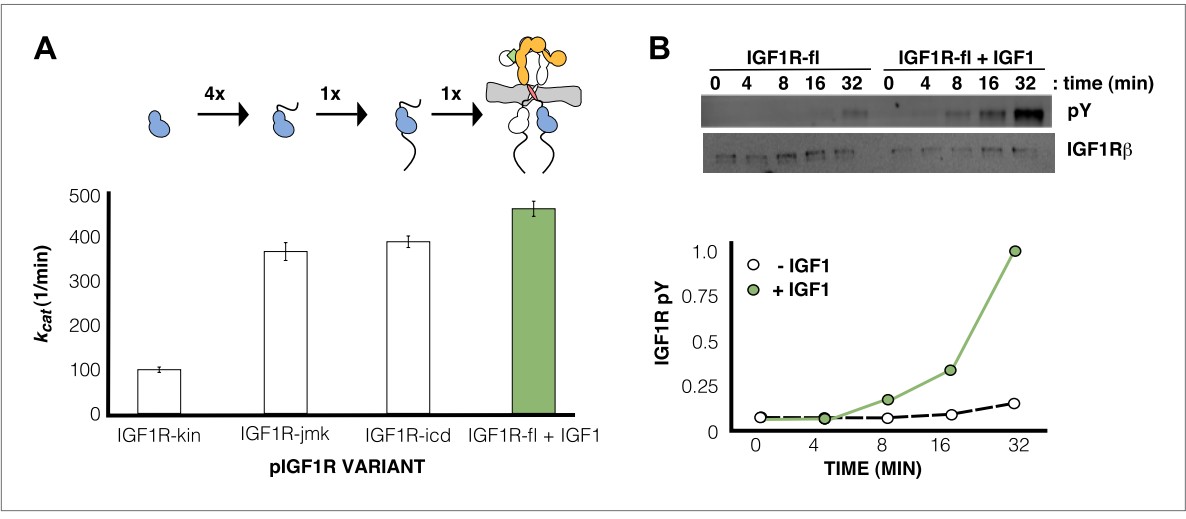

**Figure 7**. IGF1 stimulates autophosphorylation. (**A**) Bar graphs of $k_{cat}$ (min⁻¹) measurements are shown for phosphorylated forms of indicated IGF1R proteins. Diagrams of the specific fragments and the fold differences in activation are shown. (**B**) Anti-phosphotyrosine (pY) and anti-IGF1Rβ Western blots of IGF1R-fl as a function of time in the presence and absence of IGF1 (top). Quantification of band intensities in the presence (green circles) and absence (white circles) of IGF1 (bottom). See also supplemental figures.

The following figure supplements are available for figure 7:

**Figure supplement 1**. Purity of IGF1R-fl.

**Figure supplement 2**. IGF1 does not co-purify with IGF1R-fl.

**Figure supplement 3**. Analysis of phosphorylation states of IGF1R fragments.

blanket the IGF1R kinase surface were created (*Figure 8—figure supplement 2*). Sites selected for substitution are solvent exposed, conserved between IGF1R and IR, and not part of the active site. Tagged versions of all six IGF1R variants were expressed on the surface of HEK293 cells (*Figure 8A*) and their levels of phosphorylation were assessed by Western blot after immunoprecipitation. In the absence of ligand, all variant IGF1Rs showed little or no phosphorylation, similar to wild-type IGF1R, and provide no evidence for an autoinhibitory interaction between kinases (*Figure 8A*).

In contrast, two clusters of substitutions, one on the kinase N-lobe (C1) and one on the C-lobe (C4), resulted in loss of IGF1R phosphorylation in the presence of ligand (*Figure 8A*). Co-transfection of the two inactivating IGF1R N- and C-lobe variants did not restore activity indicating that they are unlikely to represent complementary surfaces of an asymmetric kinase dimer (*Figure 8B*). To ensure that the inactivating N- and C-lobe mutations did not affect the intrinsic kinase activity of the receptor, IGF1R-kin proteins bearing these mutations were expressed and purified. Both variants retained autophosphorylation and substrate kinase activities but at lower levels than wild-type IGF1R-kin (*Figure 8—figure supplement 3*), and it cannot be ruled out that this decrease in intrinsic activity underlies the loss of ligand-dependent autophosphorylation observed in cells.

## Discussion

We describe a molecular mechanism for IR/IGF1R activation in which ECD-enforced separation of TMs maintains the receptor in an inhibited state. Ligand binding relieves this inhibition by disrupting the

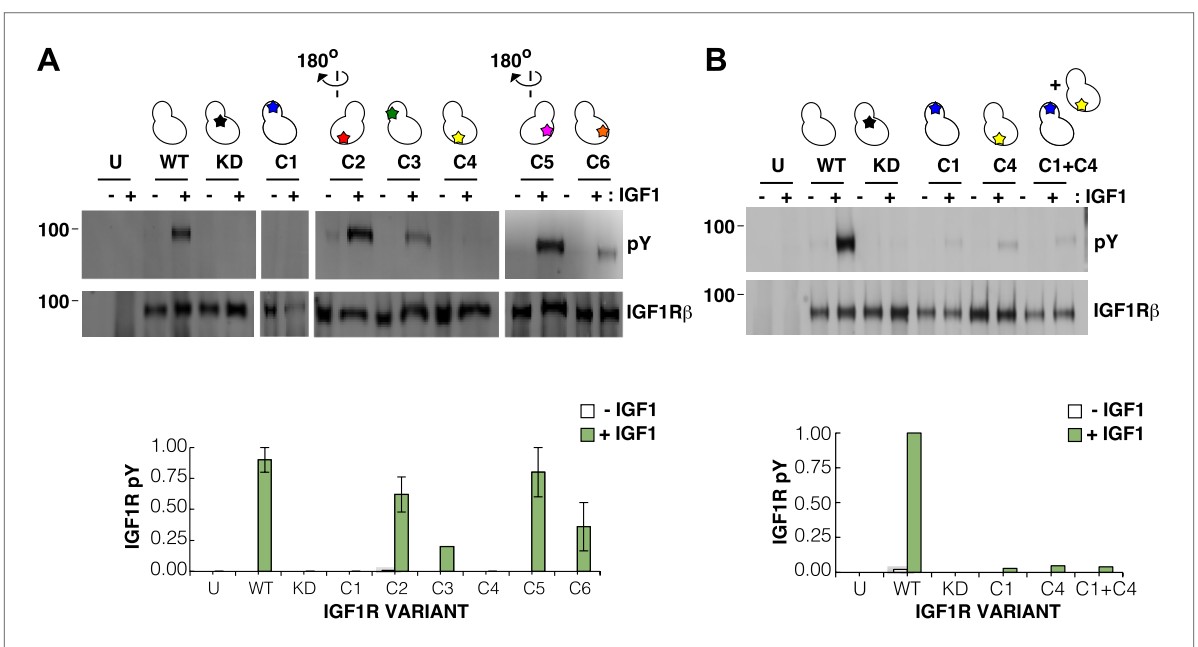

**Figure 8**. Surface mutations on the IGF1R kinase domain disrupt receptor activation. (**A**) Anti-pY and anti-IGF1Rβ Western blots of tagged IGF1R proteins immunoprecipitated from cells incubated in the presence or absence of IGF1. HEK293 cells were transiently transfected with IGF1R-fl (WT), IGF1R bearing the kinase-inactivating mutation D1105A (KD), IGF1R with clusters of surface mutations (C1-C6), or not transfected (U). Approximate locations of the mutated clusters indicated by a star on the kinase domain cartoons above each lane (top). Bar graphs showing mean receptor phosphorylation (±s.e.m.) from three independent experiments normalized to total receptor expression (for C3, only one experiment). White bars indicate cells incubated without IGF1 and green bars cells with IGF1 (bottom). (**B**) Western blots of IGF1R phosphorylation when C1 and C4 variants are co-transfected (top). Bar graph of receptor phosphorylation normalized to total receptor expression with the same coloring as in (**A**) (bottom). See also supplemental figures.

The following figure supplements are available for figure 8:

**Figure supplement 1**. Buried surface analysis of crystal lattice pairs of IR and IGF1R kinase domains.

**Figure supplement 2**. Surface analysis of IGF1R kinase domain.

**Figure supplement 3**. Analysis of the autophosphorylation of IGF1R kinase clusters.

L1:Fn2′–3′ interaction that stabilizes TM separation, freeing the TMs to associate and autophosphorylation of the kinase domains to proceed (*Figure 9A*). This model is based on a previously underappreciated dimer in the crystal structure of an Insulin bound IR ECD fragment, which provides a basis for comparing liganded and unliganded structures of the IR ECD.

To validate this model, we present FRET and mutagenesis studies that demonstrate (i) a decrease in IGF1R TM separation when ligand binds, (ii) an intrinsic propensity of the IGF1R TM and TM-icd to dimerize, (iii) that the kinase is active in this associated state, consistent with the increased activity of the IGF1R ICD when fused to a dimeric partner (*Baer et al., 2001*), (iv) that the role of the ECD is primarily to inhibit this intrinsic kinase activity in the absence of ligand, rather than promote a specific active state, (v) that IGF1R TMs associate in an active form of the full-length receptor, and (vi) that IGF1R samples the active state even in the absence of ligand. Enzymatic studies of full-length IGF1R and IGF1R fragments further demonstrate no role for allostery in maintaining IGF1R activity once it is phosphorylated, indicating that the key step regulated by ligand binding is autophosphorylation. Although ligand binding has no effect on the intrinsic activity of phosphorylated IGF1R, the short half-life of IGF1R phosphorylation when phosphatases are present suggests the continued presence of ligand may be needed to maintain IGF1R activity (*Kleiman et al., 2011*). This efficiency of cellular

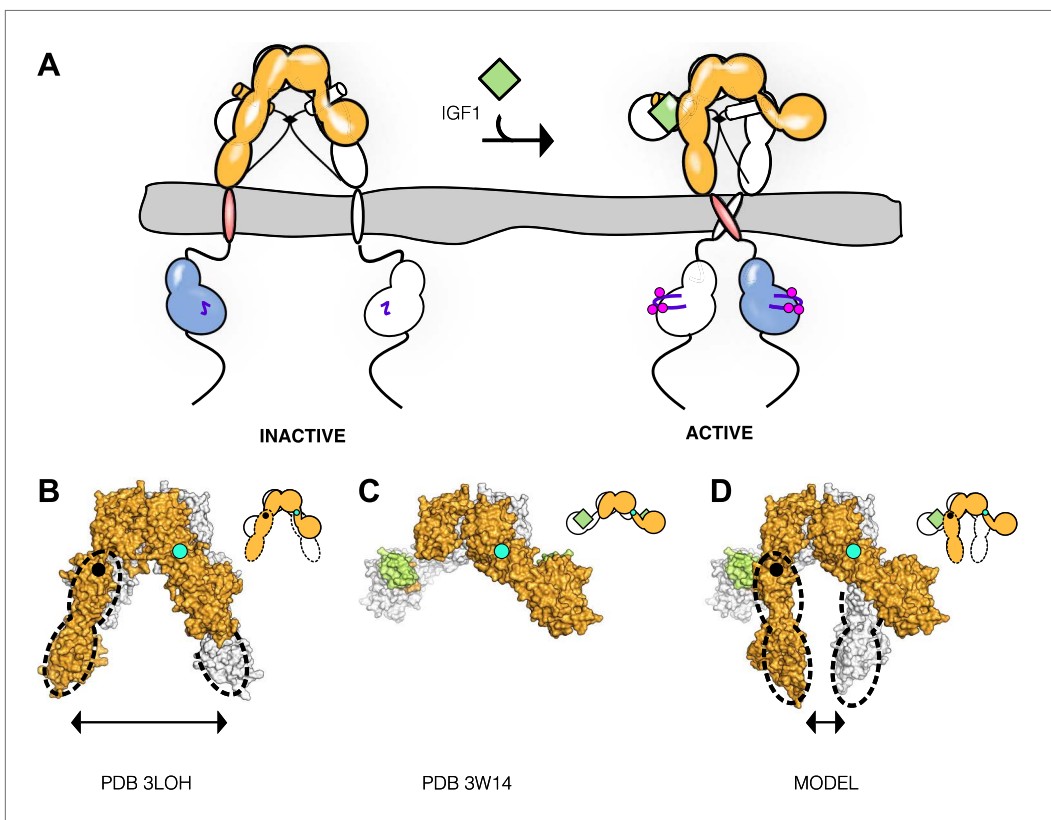

**Figure 9**. Model of IGF1R activation. (**A**) Cartoon model of IR family activation. The IDs are shown as black lines in the ECDs, the disulfide linkages as black diamonds, phosphorylation as pink circles, activation loops as purple lines, ligand as green diamonds, and αCTs as cylinders. Surface representations and corresponding cartoon diagrams of (**B**) the ECD of unliganded-IR (PDB 3LOH; *Smith et al., 2010*), (**C**) insulin bound to the IR ECD fragment (PDB 3W14; *Menting et al., 2013*), or (**D**) a model of the entire IR ECD bound to two Insulin molecules. For each panel, one αβ subunit is colored white and the other orange, and Insulin green. Dashed lines outline the Fn2-3 domain tandems. The hinge point between F1 and Fn2-3 domains is indicated by black circles and cyan circles indicate the hinge point between CR and L2 domains.

The following figure supplement is available for figure 9:

**Figure supplement 1**. Linkage of αCTs contributes to negative cooperativity through destabilization of the second L1′:Fn2′-3′ site following ligand binding.

phosphatases may also explain why transient sampling of the active state by IGF1R does not lead to constitutive activity.

Previous models for IR activation have posited only modest changes in receptor conformation when ligand binds and that 'twist' rather than lateral movement of the Fn2–3 domains is the activating signal or that the TM regions move from associated to apart during IR activation (*Ward et al., 2013*; *Lee et al., 2014*). We show here that ligand binding to IGF1R leads to TM association. Thus the Fn2–3 domains must undergo a large movement upon ligand binding to accommodate the new position of the TMs. To model this movement, we appended the Fn2–3 tandems from the unliganded IR ECD structure to the structure of the IR ECD fragment bound to Insulin. The Fn2–3 tandems were then manually repositioned using only the constraints that the ligand bound position of these 'legs' must be compatible with the TM dimer observed in MD simulations and that the residues on Fn2 that compose Site 2 contact ligand. Assuming the connection between Fn1 and Fn2 to be flexible (it buries ~500 $Å^2$ of surface area) and the Fn2–3 tandem to be rigid, we find it possible to place the Fn2–3 tandems so that Site 2 residues appose ligand and that the ECD C-termini are compatible with TM dimerization (*Figure 9B–D*).

We have modeled IR/IGF1R with two bound ligands (*Figure 9D*) but receptors with a single ligand bound may represent an important active state. Negative cooperativity implies that ligand binding at one site influences assembly of the second site. The most likely conduit of information between sites is through the αCTs, which are connected by an inter-subunit disulfide bond 8 residues from their N-termini. The αCTs are 30 Å farther apart in the liganded IR structure than in the unliganded IR structure (*Smith et al., 2010*; *Menting et al., 2013*). This increased separation is only possible because the inter-subunit disulfides that link the αCTs were deleted in the crystallized fragment of IR. In an intact receptor, the altered position of αCT following binding of one ligand would restrict the reach of the opposite αCT, likely hindering assembly of the second binding site and contributing to negative cooperativity.

The intrinsic propensity of the TMs to associate may also contribute to negative cooperativity. Disruption of the L1:Fn2'–3' interaction by ligand binding removes a restraint on the relative position of the TMs that served as a barrier to TM association (*Figure 3*). Decreasing this barrier may shift the energetic balance between maintenance of the L1':Fn2–3 interaction at the unliganded site and the intrinsic propensity of the TMs to associate and couple TM association to rearrangement of elements of the unliganded binding site (*Figure 9—figure supplement 1*). This mechanism may underlie the restoration of negative cooperativity of soluble IR ECDs when their C-termini are artificially tethered (*Hoyne et al., 2000*).

The results discussed so far implicate ECD-enforced TM separation as a key feature maintaining IGF1R and IR inactive states. The question then arises of how TM separation maintains the kinases in an inactive state. The simplest explanation is that spatial separation is sufficient to preclude one kinase from phosphorylating the other. The N-termini of subunits of an IGF1R kinase dimer captured in an apparent *trans* phosphorylation state are ~50 Å apart (*Wu et al., 2008*), however, and the 27 residues separating each kinase domain from its TM could easily span the distance needed to reach the ~120 Å separation imposed by the ECD. The IGF1R intracellular juxtamembrane regions must thus adopt some structure in the inactive state to preclude autophosphorylation, but secondary structure prediction algorithms only identify a short 7 amino-acid β strand in this region. Outside of three positively charged residues immediately following the TM, no conserved regions of positive charge comparable to the region in EGFR thought to mediate interactions with the membrane are present (*Jura et al., 2009*). Conversely, the fourfold enhancement of kinase activity when the juxtamembrane region is present implies some structure in this region in the active state. Attempts to identify regions of the IGF1R kinase important for maintaining the inactive state by either mutagenesis or inspection of kinase crystal lattices for a repeated interaction proved unsuccessful, however. No IGF1R kinase surface mutations led to higher basal activity, and no repeated interactions or regions of positive charge are apparent in IGF1R kinase crystals that could mediate self interactions or interactions with the plasma membrane as has been proposed for EGFR (*Arkhipov et al., 2013*).

In contrast, the absence of allosteric enhancement of the activity of phosphorylated IGF1R implies the absence of a stable interaction between kinase domains in the active state. Consistent with this conclusion, surface mutagenesis of the IGF1R kinase demonstrates the absence of an EGFR-like asymmetric kinase dimer. Absence of allostery in the active state may be a general feature of RTKs whose activity is enhanced by multiple activation loop phosphorylations, including FGFR, Met, VEGFR, Alk,

MUSK, Kit, and their homologs (*Lemmon and Schlessinger, 2010*). Transient intermediate states may direct the order of tyrosines phosphorylated, as observed for FGFR (*Lew et al., 2009*), but once fully phosphorylated we suspect the activity of these kinases will also prove independent of allosteric stimulation. This behavior contrasts with that of EGFR, which relies on allostery via a kinase asymmetric dimer for activation rather than phosphorylation of the activation loop (*Zhang et al., 2006*).

Although IGF1R ICDs do not appear to form a stable interaction in the active state, TM dimerization does appear to occur upon ligand binding. This observation is consistent with previous studies demonstrating a coupling between ligand binding and association of IR ECD C-termini as well as results with chimeric receptors. When the IR ECD is expressed in soluble form it binds with nearly 1000-fold lower affinity than full-length IR, but when the C-termini of the ECD are fused to either a leucine zipper (*Hoyne et al., 2000*) or an immunoglobulin Fc region (*Bass et al., 1996*), high-affinity, native-like ligand binding is restored (*Schaffer, 1994*). TM dimerization is thus not only consistent with high-affinity ligand binding but promotes it. Furthermore, a chimeric receptor with an IR ECD and an EGFR TM and ICD can be activated by Insulin (*Riedel et al., 1986*). As EGFR activation involves TM dimerization (*Endres et al., 2013*), the Insulin-bound IR ECD is also compatible with TM dimerization.

We present here a new molecular model for regulation of IGF1R and IR activity. The key feature of this model is maintenance of an inactive state by ECD-enforced separation of the TMs in the absence of ligand. Ligand binding releases this constraint, resulting in TM association and unleashing of the intrinsic propensity of the kinase regions to autophosphorylate and activate. This model is consistent with recent crystal structures of the IR ECD and suggests that some structure in the intracellular region precludes autophosphorylation in the unliganded state of the receptor. Conversely, enzymatic studies show that once phosphorylated and active, no structure outside of the kinase and juxtamembrane regions is needed to stimulate full kinase activity. This model provides a simple molecular context for understanding several features of IR/IGF1R activation and suggests directions for future study. It also suggests that chimeric proteins with IR or IGF1R ECDs may prove valuable for assessing the role of TM separation in other signaling systems or as ligand-dependent activity switches when fused, for example, to split enzymes.

## Materials and methods

### Expression and purification of the IGF1R kinase domain (IGF1R-kin)

A recombinant baculovirus was engineered to encode residues 956–1256 of IGF1R with the substitution C1207S using the pFastBacHT B plasmid (Life Technologies, Carlsbad, California). Sf9 cells at a concentration of $2 \times 10^6$ cells/ml were infected with this P3 baculovirus and harvested 3 days post-infection. The purification protocol was based on a previously published procedure (*Favelyukis et al., 2001*). Cells were lysed by using a French pressure cell in buffer containing 50 mM Tris pH 8, 50 mM NaCl, 0.2% Triton X-100, Benzonase nuclease (Sigma, St. Louis, Missouri), and protease inhibitor tablets (Roche, Germany). IGF1R-kin was isolated by immobilized metal affinity chromatography (IMAC), and the histidine-tag was removed by overnight incubation with TEV protease. Calf-intestinal phosphatase (CIP) (New England Biolabs, Ipswich, Massachusetts) was also included in the incubation to dephosphorylate the protein. The next day, protein was loaded onto a size-exclusion column (Superdex 75 26/60) equilibrated in 10 mM Hepes pH 7.5, 150 mM NaCl, and 5 mM β-ME, and the monomeric fractions pooled. Pooled fraction were then loaded onto a monoQ column (5/50 GL) equilibrated in 10 mM Hepes pH 7.5, 5 mM β-ME and eluted with a linear gradient of NaCl. Fractions corresponding to the unphosphorylated protein, as judged by Coommassie Brilliant Blue stained native gel, were pooled, concentrated, and flash frozen in 20% glycerol. Phosphorylated fractions were pooled, concentrated, and incubated with 10 mM ATP and 30 mM MgCl$_2$, desalted, and then repurified on a mono Q column. Fractions corresponding to the fully phosphorylated IGF1R-kin, as judged by Coommassie stained native gel, were pooled, concentrated, and flash frozen in 20% glycerol.

### Expression and purification of the IGF1R juxtamembrane kinase region (IGF1R-jmk)

IGF1R-jmk, residues 930–1256 of IGF1R with C1207S substitution, was expressed in Sf9 cells as described for IGF1R-kin. The purification protocol was based on a previously published method (*Craddock et al., 2007*). Cells were lysed using a French pressure cell in buffer containing 50 mM Tris pH 8.0, 1 M KCl, 10% glycerol, 0.2% Triton X-100, Benzonase, and protease inhibitor tablets. Clarified

lysate was applied to an IMAC column (Biorad, Hercules, California) and eluted with imidazole. IGF1R-jmk was then incubated overnight with TEV and CIP to remove phosphorylation and the N-terminal histidine tag and dialyzed into 10 mM Tris pH 8.0, 200 mM KCl, 10% glycerol, 0.2% Triton-X100, 2.5 mM β-ME, 1 mM PMSF at 4°C. The cleavage reaction was reapplied to an IMAC column and the flow-through collected. The flow through was diluted to a final concentration of 50 mM KCl and applied to a HiTrap Q (GE Lifescience, Pittsburgh, Pennsylvania) column equilibrated in 10 mM Hepes pH 7.5, 0.2% Triton X-100, and 5 mM β-ME and eluted with a linear gradient of KCl. Fractions containing IGF1R-jmk were pooled, diluted to a final concentration of 50 mM KCl, and loaded onto a monoQ column equilibrated in the same buffer as for the HiTrap Q column. IGF1R-jmk was eluted with a gradient of KCl. Fractions containing unphosphorylated IGF1R-jmk were analyzed as described above, concentrated, and flash frozen in 20% glycerol. Phosphorylated IGF1R-jmk fractions were pooled, concentrated, and incubated with 10 mM ATP, 30 mM MgCl$_2$, and 10% glycerol. The sample was then desalted and purified via monoQ to isolate the fully phosphorylated IGF1R-jmk, which was then concentrated and flash frozen as described previously.

### Expression and purification of the IGF1R intracellular region (IGF1R-icd)

A recombinant baculovirus was engineered to encode the entire IGF1R intracellular region, residues 930–1337 with the substitution C1207S and a C-terminal SBP tag, using the pFastBacHT B plasmid (Life Technologies). Expression, lysis, IMAC, and cleavage were the same as for IGF1R-jmk. Histidine-tagged *Yersinia enterocolitica* phosphatase (YOP) was used for dephosphorylation. After cleavage, IGF1R-icd was loaded onto an IMAC column and the flow-through was collected. Untagged IGF1R-icd was then loaded onto a Streptactin HiTrap column (GE Lifescience) and eluted with 10 mM Hepes pH 7.5, 150 mM KCl, 5 mM β-ME, and 2.5 mM desthiobiotin. This purification steps ensures that IGF1R-icd had an intact C-terminal tail. IGF1R-icd was then purified on a G75 size-exclusion column equilibrated in the same buffer lacking desthiobiotin. Monomeric IGF1R-icd was then pooled, diluted, and purified as described above to isolate unphosphorylated IGF1R-icd. To generate fully phosphorylated IGF1R-icd, partially phosphorylated IGF1R-icd was incubated with 10 mM ATP, 30 mM MgCl$_2$, 0.05% Triton X-100, and 10% glycerol for 4 hr at 4°C. This sample was then desalted into 10 mM Tris pH 8, 150 mM NaCl, 0.05% Triton X-100, concentrated, supplemented with 20% glycerol and flash-frozen.

### Expression and purification of full-length IGF1R (IGF1R-fl)

IGF1R-fl, residues 1–1337 with a C-terminal histidine tag, was stably expressed in HEK293 GnTi⁻ (ATCC, Mannasas, Virginia) cells by Fugene transfection followed by FACS sorting. IGF1R-fl expressing cells were grown in a shaking incubator supplemented with 8% CO$_2$ in Freestyle Medium (Invitrogen, Carlsbad, California) supplemented with 1% FBS and 2 mM Glutamine, pelleted, and lysed in a liquid nitrogen mill (Spex Sample Prep, Metuchen, New Jersey). Lysed cells were resuspended in 50 mm Tris pH 8.0, 100 mM NaCl, 5 mM EDTA, 1 mM PMSF, 10% glycerol, and 2% β-octylglucoside for 1 hr at 4°C and spun at 200,000×*g* at 4°C for 1 hr. Clarified lysate was incubated with cyanogen bromide-resin (GE Healthcare) coupled to the monoclonal antibody 24–55 (*Soos et al., 1992*) for 1 hr. The resin was separated by centrifugation, washed sequentially in buffer containing 1 M NaCl, 0.5 M Urea, or 10 mM MgCl$_2$. Protein was eluted with 25 mM Tris pH 8.0, 200 mM NaCl, 1 M MgCl$_2$, 10% glycerol, and 2% β-octylglucoside, buffer exchanged on a desalting column into 20 mM Tris pH 8.0, 100 mM NaCl, 1 mM EDTA, 0.1 mM PMSF, 10% glycerol, and 2% β-octylglucoside, and loaded onto a monoQ column equilibrated in 10 mM Tris pH 8.0, 1 mM EDTA, 0.1 mM PMSF, 2.5% glycerol, and 0.03% Triton X-100. IGF1R-fl was eluted with 300 mM NaCl, and IGF1R-fl containing fractions pooled, concentrated in a Pierce 100 kDa molecular weight cut-off spin concentrator, supplemented with 20% glycerol, and flash frozen. To produce dephosphorylated IGF1R-fl, the above protocol was followed with the addition of YOP to the resuspension step. To produce phosphorylated protein, the above protocol was followed with the addition of 30 mM MgCl$_2$ and 10 mM ATP to the cell resuspension step.

### Expression and purification of IGF1

A plasmid encoding an N-terminal histidine tagged version of IGF1 was transformed into BL21(DE3) Rosetta2 cells (EMD-Millipore, Billerica, Massachusetts). Cells were grown at 37°C in Terrific Broth to an OD600 of 0.4, induced with 1 mM IPTG, and harvested 2 hr after induction. Cells were lysed using a French pressure cell in 20 mM Tris pH 8.0 and 6 M Urea. Clarified supernatants were loaded onto an IMAC column and eluted into 12.5 mM Tris pH 8.0, 3 M Urea, and 125 mM imidazole. Elution fractions containing IGF1 were pooled and dialyzed three times: first for 2 hr at room temperature against

20 mM Glycine pH 10.5, 15 mM DTT, and 2 M Urea, second for 18 hr against 20 mM Glycine pH 10.5, 3 mM DTT, 1 M Urea, 1 M NaCl, 0.5 µM $CuCl_2$, and 20% Ethanol, and third for 2 hr at 4°C against 25 mM Tris pH 8, 200 mM NaCl. The sample was centrifuged to remove insoluble aggregates, and the supernatant applied to a G75 sizing column equilibrated in 5 mM Tris pH 8, 200 mM NaCl. Fractions containing monomeric IGF1 were pooled, concentrated, and flash frozen in 50% glycerol.

## Purification of the 24–55 anti-IGF1R antibody

A hybridoma cell line expressing monoclonal antibody 24–55 (*Soos et al., 1992*) (a generous gift from K Siddle) was grown in a shaking incubator in hybridoma medium (Gibco, Carlsbad, California). Conditioned medium was applied to a Protein G column, and the antibody eluted with 100 mM Glycine pH 2.7 and neutralized with Tris pH 9.0. The 24–55 antibody was concentrated and dialyzed into 100 mM $NaHCO_3$ and 500 mM NaCl and coupled to cyanogen bromide-resin according to manufacturer's directions.

## Expression and purification of fluorescent proteins

Plasmids encoding N-terminally hexa-histidine tagged EYFP, mCherry, or mTurquoise were transformed into *Escherichia coli* BL21(DE3) cells. Cultures were grown in Luria Broth at 37°C to $OD_{600}$ ~0.6, and the temperature dropped to 18°C for 30 min. IPTG was added to a final concentration of 1 mM, and the culture was grown overnight (~16 hr). Cells were harvested by centrifugation and lysed using a French pressure cell. An IMAC column was loaded with clarified lysate, washed with 20 mM imidazole, and eluted with 250 mM imidazole. Protein-containing fractions were pooled, dialyzed, concentrated, and flash frozen. To minimize photobleaching, all expression and purification steps were performed in the dark. The purity of the fluorescent proteins was >90% after the nickel column as judged by Coommassie Brilliant Blue stained SDS-PAGE.

## In vitro kinase assays

Radiometric kinase assays to determine kinetic parameters were performed as described previously (*Qiu et al., 2009*). Reactions were performed in 25 µl of a buffer containing 50 mM HEPES pH 7.5, 50 mM NaCl, 10% Glycerol, 0.5 mM DTT, 0.2% Triton X-100, 10 mM $MgCl_2$, 0.2 mg/ml ovalbumin, 0.1 mM $Na_3VO_4$, and either fixed or varying concentrations of ATP and Biotin-KKEEEEYMMMMG as the peptide substrate. A previously identified consensus sequence was used as the peptide substrate with an extra N-terminal lysine added to help with solubility (*Songyang et al., 1995*). Reactions were performed at 30°C and quenched by the addition of 10 µl of 100 mM EDTA. 10 µl of 20 mg/ml avidin (ThermoFisher, Waltham, Massachusetts) was added to the reactions and the mixture was transferred to a concentrator with a 30-kDa molecular weight cutoff (Pall, Port Washington, New Jersey). Samples were then washed three times in buffer containing 0.5 M NaCl and 0.5 M $Na_2HPO_4$, pH 8.3 and counted. Turnover of the limiting substrate was less than 10%. Reactions were performed in duplicate, and values were generally within 20%. Data were fit to a non-linear curve using the Michaelis–Menten equation to obtain apparent $K_m$ and $k_{cat}$ values. Concentrations of ATP and peptide used are listed in the table below.

|  | pY | Experiment | Concentration (µM) |
|---|---|---|---|
| IGF1R–fl + IGF1 | + | $K_m$ ATP | 500, 400, 300, 250, 125, 62.5, 31.3, 15.6, 7.8 |
| IGF1R–fl + IGF1 | + | $K_m$ Peptide | 600, 300, 150, 75, 37.5, 18.8, 9.4 |
| IGF1R–fl + IGF1 | − | $K_m$ ATP | 2000, 1000, 500, 250, 125, 62.5, 31.3, 15.6, 7.8 |
| IGF1R–fl + IGF1 | − | $K_m$ Peptide | 500, 250, 125, 62.5, 31.3, 15.6, 7.8, 3.9 |
| IGF1R–fl | + | $K_m$ ATP | 500, 400, 300, 250, 125, 62.5, 31.3, 15.6, 7.8 |
| IGF1R–fl | + | $K_m$ Peptide | 500, 400, 250, 125, 62.5, 31.3, 15.6, 7.8 |
| IGF1R–fl | − | $K_m$ ATP | 1000, 500, 250, 125, 62.5, 31.3, 15.6, 7.8 |
| IGF1R–fl | − | $K_m$ Peptide | 1000, 500, 250, 125, 62.5, 31.3, 15.6 |
| IGF1R–icd | + | $K_m$ ATP | 500, 250, 125, 62.5, 31.3, 15.6, 7.8, 3.9 |
| IGF1R–icd | + | $K_m$ Peptide | 1000, 500, 250, 125, 62.5, 31.3, 15.6, 7.8 |
| IGF1R–icd | − | $K_m$ ATP | 1000, 500, 250, 125, 62.5, 31.3 |
| IGF1R–icd | − | $K_m$ Peptide | 1000, 500, 250, 125, 62.5, 31.3 |

*Table Continued on next page*

Table Continued

| | pY | Experiment | Concentration (µM) |
|---|---|---|---|
| IGF1R-jmk | + | $K_m$ ATP | 500, 250, 125, 62.5, 31.3, 15.6 |
| IGF1R-jmk | + | $K_m$ Peptide | 300, 150, 75, 37.5, 18.8, 9.4, 4.7 |
| IGF1R-jmk | − | $K_m$ ATP | 1000, 250, 125, 62.5, 31.3 |
| IGF1R-jmk | − | $K_m$ Peptide | 625, 312.5, 156.3, 78.1, 39.1 |
| IGF1R-kin | + | $K_m$ ATP | 1000, 500, 250, 125, 62.5, 31.3 |
| IGF1R-kin | + | $K_m$ Peptide | 375, 187.5, 93.8, 46.9, 23.4, 11.7, 5.9 |
| IGF1R-kin | − | $K_m$ ATP | 1000, 750, 500, 375, 250, 125, 62.5 |
| IGF1R-kin | − | $K_m$ Peptide | 1250, 625, 312.5, 156.3, 78.1, 39.1 |

The linear range of activity for the kinase with respect to either time or concentration was established using multiple time points. The reaction buffer contained 300 µM peptide and 500 µM ATP. Reaction times ranged between 2 and 30 min and enzyme concentrations ranged from 0.3 nM to 200 nM.

## In vitro autophosphorylation assays

10 nM of purified IGF1R-fl was incubated with 2 mM ATP in the same buffer as the in vitro kinase assays either without or without 50 nM IGF1. Upon addition of ATP, the reactions were incubated at 30°C, and samples were removed at various time points and immediately quenched with 83 µM EDTA. Samples were separated by denaturing SDS-PAGE electrophoresis and transferred to PVDF membranes. Blots were probed with an anti-phosphotyrosine (pY) (4G10, EMD-Millipore) or anti-IGF1Rβ antibodies, scanned with a Typhoon Scanner (GE Lifesciences), and intensities quantified with ImageJ. For autophosphorylation of IGF1R-kin variants, the same assay was used except IGF1R proteins were at 1 µM and ATP at 1 mM.

## Cell-based IGF1R activity assays

IGF1R cell-based assays were based on established procedures (*Liu et al., 2012*). HEK293 cells (ATCC) were maintained in adherent culture in DMEM:F12 supplemented with 5% FBS. HEK293 cells were chosen due to low background endogenous IGF1R as judged by Western blot and equal transfection efficiency of IGF1R wild-type and kinase-inactive (D1205A/N) when compared with transient transfection of *igf1r* (−/−) mouse embryonic fibroblasts (a generous gift from R Baserga). HEK293 cells were plated in a six-well plate at $1 \times 10^6$ cells/well and transiently transfected with IGF1R expression plasmids using polyethylenimine (PEI, linear MW 25,000; Polysciences, Inc., Warrington, Pennsylvania) at an optimized ratio of 1 µg DNA: 3 µg PEI. Variant IGF1R genes were subcloned into pSSX-F, a version of pSGHV0 modified to eliminate the growth hormone tag and add a C-terminal Flag tag (*Leahy et al., 2000*). After 18 hr, cells were washed three times with 2 ml Ham's F12 supplemented with 1 mg/ml BSA and incubated in this medium for 3 hr at 37°C to serum starve. In designated wells, 20 nM IGF1 was added and incubated for 30 min at 37°C.

For cross-linking assays and assays using truncated IGF1R variants, the wells were washed with ice-cold phosphate buffered saline and then lysed for 30 min at 4°C in 250 µl of RIPA buffer (50 mM Tris pH 8, 150 mM NaCl, 1% NP-40, 0.5% sodium deoxycholate, and 0.1% SDS) supplemented with 1 mM activated $Na_3VO_4$, 1 mM PMSF, Benzonase nuclease (Sigma), and 10 mM iodoacetamide to prevent further disulphide-bond formation during lysis (*Cao et al., 1992*). Lysates were clarified and samples were separated by either denaturing reducing or non-reducing SDS-PAGE electrophoresis. Western blot analysis was performed using anti-IGF1Rβ (Santa Cruz Biotechnologies, Dallas, Texas) Western blots were developed using ECL2 (Thermo Scientific) and scanned using a Typhoon Imager. Bands corresponding to phosphorylated IGF1R from the 4G10 Western blot were quantified using ImageJ and normalized to receptor expression from quantified bands of the IGF1R-β Western blot for each experiment.

For autophosphorylation analysis, the wells were washed with ice-cold phosphate buffered saline and lysed in supplemented RIPA buffer lacking iodoacetamide. The cell suspension was clarified, and total protein concentration determined using the BCA protein assay (Thermo Scientific-Pierce) to normalize cell lysates. For assays with truncated IGF1R proteins, Western blot analysis was performed on cell lysates using anti-IGF1Rβ or anti-IGF1R phosphotyrosine 1135 (pY1135) (Cell Signaling Technology,

Danvers, Massachusetts) antibodies. For assays with IGF1R-fl proteins with kinase clusters, single cysteine substitutions, or TM mutations immunoprecipitation was performed. Anti-Flag-M2 (Sigma) was added at 0.5 µg/ml to lysate followed by the addition of 20 µl Protein G Sepharose 4 Fast Flow (GE Healthcare). Lysates were then incubated overnight at 4°C. Beads were then washed three times with 1 ml of RIPA buffer supplemented with 1 mM activated $Na_3VO_4$. Beads were eluted by the addition of 20 µl of 5× LDS loading buffer containing 10% fresh β-mercaptoethanol. Equal amounts of eluted proteins were analyzed as described for the cross-linking analysis but using both anti-pY and anti-IGF1Rβ antibodies.

## Two photon FRET microscopy on living cells

We employed two donor–acceptor FRET pairs for this study: mTurquoise-YFP and YFP-mCherry. We calculated the Förster radius for each pair to be 54.5 Å (mTurquoise-EYFP) and 53.1 Å (EYFP-mCherry), making them suitable for investigating conformational changes on the order of 30–90 Å. CHO cells were cultured in 35-mm collagen-coated glass bottomed dishes (MatTek Corporation, Ashland, Massachusetts) using phenol red-free DMEM-F12 supplemented with 5% FBS and 1 mM *L*-glutamine. Plasmids encoding the IGF1R extracellular and transmembrane regions fused after GGSGGS to mTurquoise (FRET donor) or EYFP (FRET acceptor) were co-transfected using Fugene HD (Promega, Madison, Wisconsin) at a mass ratio of 3:1 (Fugene:DNA), according to the manufacturer's protocol. Transfection proceeded for 24 hr at 37°C, 5% $CO_2$.

We used the OptiMiS TruLine Spectral Scanning System (Aurora Spectral Technologies, Milwaukee, Wisconsin) for two-photon microscopy. A solid-state continuous wave laser pumped a mode-locked Ti:Sapphire laser (MaiTai DeepSee, Newport Corporation, Irvine, California) that generated near infrared pulses in the wavelength range of 800–960 nm ($\lambda_{exc}$ for mTurquoise = 800 nm; $\lambda_{exc}$ for eYFP = 960 nm). The beam was focused using a 63× objective (Nikon, Japan). Fluorescence emission from the sample chamber was projected through a transmission grating onto a cooled electron-multiplying charge-coupled device. The full spectral scans of all pixels in the viewing field (300 × 440 pixels) were completed in about 10 s.

Image acquisition and processing are based on the method described by *Raicu et al. (2008)*. Prior to imaging, CHO cells were washed three times in 1× PBS to remove all traces of FBS, then serum starved in phenol red-free DMEM-F12 without FBS. After 5 hr of starvation, the media were changed a final time and the cells were imaged directly in the two-photon microscope. Individual donor and acceptor fluorescence spectra were collected from cells expressing either the donor- or acceptor-fused IGF1R alone. A single 'FRET' scan ($\lambda_{exc, two photon}$ = 800 nm) was acquired for each cell that coexpressed both EYFP and mTurquoise-fusion proteins. The full spectral emission profile was collected from 450–600 nm (1 nm resolution), and the FRET efficiency per pixel ($E_{app}^D$) was calculated using a Matlab program, according to the equations described (*Raicu et al., 2008*).

$$E_{app}^{Dq} \equiv \frac{F^D(RET)}{F^D(\lambda_{ex})} = \frac{1}{1 + \dfrac{Q^A}{Q^D} \dfrac{k^{DA}}{k^{AD}} \dfrac{w^D}{w^A}}$$

where $F^D(RET)$ and $F^D(\lambda_{ex})$ equal the fluorescence emission of the donor in the presence and absence of resonance energy transfer, respectively. $Q^A$ and $Q^D$ are the quantum yields of the donor an acceptor fluorophore, and $k^{DA}$ and $k^{AD}$ are the maximum emission intensities of the donor in the presence of the acceptor, and the acceptor in the presence of the donor, respectively. The integrals of the individual emission spectra for the donor and acceptor are given by $w^D$ and $w^A$. Representative pixels in each cell membrane were examined by eye to ensure a good fit to the FRET model. We calculated the Förster radius for the mTurquoise-EYFP donor–acceptor pair to be 54.5 Å using measured absorption and emission spectra from purified proteins (*King et al., 2014*).

## Confocal microscopy and quantitative imaging FRET

CHO cells were cultured in DMEM-F12 supplemented with 5% FBS and 1 mM *L*-glutamine. Cells were seeded in 35-mm dishes at a density of $4 \times 10^4$ cells per well and grown for 24 hr at 37°C in 5% $CO_2$, then transiently co-transfected with plasmids encoding IGF1R TM-EYFP (3 µg) and IGF1R TM-mCherry (6 µg) using the Fugene HD transfection reagent at a mass ratio of 3:1 (Fugene:DNA) according to the manufacturer's protocol.

After 24 hr, cells were washed twice with 1 ml of 30% PBS, one minute per wash. 1 ml of chloride salt vesiculation buffer (*Del Piccolo et al., 2012*) was added to each well, and the wells were incubated at 37°C, 5% $CO_2$. Vesiculation buffer is composed of 200 mM NaCl, 5 mM KCl, 0.5 mM $MgCl_2$, 0.75 mM $CaCl_2$, and 100 mM bicine, pH 8.5. Vesiculation reached completion after about 12 hr, and the entire well supernatant was transferred to 4-well chambered slides (Thermo Scientific, Nunc Lab-Tek II). The wells were allowed to equilibrate to room temperature before imaging.

Vesicle images were acquired using a Nikon C1 laser scanning confocal microscope equipped with a 60× water immersion objective. Three scans were taken for each vesicle: (i) a 'donor' scan ($\lambda_{exc}$ = 488 nm, $\lambda_{em}$ = 500–530 nm), (ii) an 'acceptor' scan ($\lambda_{exc}$ = 543 nm, $\lambda_{em}$ = 650 nm longpass), and (iii) a 'FRET' scan ($\lambda_{exc}$ = 488 nm, $\lambda_{em}$ = 565–615 nm). The donor and FRET scans used a 488 nm argon ion laser excitation source, while the acceptor scan used a 543 nm He-Ne laser. The image resolution was 512 × 512 pixels, with a pixel dwell time of 1.68 μs. The gains were set to 7. All images were processed using a Matlab program developed in the Hristova laboratory. The program finds the boundary of each vesicle, verifies that the vesicle is present in all three scans, and fits the intensity profile across the membrane to a Gaussian function. The baseline is fitted with an error function. Details for the calculations of FRET efficiency are provided in the study by *Chen et al. (2010a, 2010b)*. We calculated the intrinsic FRET efficiency as:

$$\text{Intrinsic FRET} = (E_{app} - E_{proximity})/x_a$$

where $E_{app}$ is the apparent FRET efficiency observed in each vesicle, $E_{proximity}$ is contribution of FRET from nonspecific interactions (*King et al., 2014*), and $x_a$ is a correction factor which accounts for varying ratios of donor and acceptor molecules within each vesicle.

To determine fluorescent protein concentration in the vesicles, vesicle fluorescence intensities were normalized to standard solutions of mCherry and EYFP purified from *E. coli* BL21 cells using the formula (*Chen et al., 2010a*, *2010b*):

$$E + 1 - \frac{I_D^m}{I_D^m + I_{D,corr}^m}$$

where $I_D^m$ is the emission intensity of the donor per unit area of the membrane in the presence of RET, and $I_{D,corr}^m$ is the donor emission intensity in the absence of RET (*Chen et al., 2010a*, *2010b*). Bleed-through coefficients were calculated from confocal images of standard solutions of purified EYFP and mCherry. Using absorption and emission spectra of purified proteins, we calculated the Förster radius for the EYFP-mCherry donor–acceptor pair to be 53.1 Å.

## Molecular dynamics simulations

A simulation system was set up by tiling an equilibrated 15% POPS–85% POPC bilayer structure to an orthorhombic box of desired dimensions. The simulation box was filled with water molecules and 0.15 M NaCl. The POPS fraction was chosen to mimic the abundance of anionic lipids in mammalian plasma membrane (*Zachowski, 1993*; *van Meer et al., 2008*), in which ~10% of lipids are PS species, and few more percent are other anionic species, such as phosphoinositides. The bilayer was modeled with POPS only in the intracellular leaflet. The IGF1R TM helices were embedded in the bilayer. The simulation box of two IGF1R TM helices for the two simulations of 12 μs and 34 μs is of 65 Å × 65 Å × 86 Å in dimensions and the simulation box of four IGF-1R transmembrane helices for the other two simulations of 24 μs and 36 μs is of 81 Å × 81 Å × 93 Å in dimensions.

The systems were parameterized using the CHARMM36 forcefield for lipids and salt (*Beglov and Roux, 1994*), CHARMM TIP3P for water molecules (*Neria et al., 1996*), and the CHARMM22* forcefield (*Mackerell et al., 1998*, *2004*; *Piana et al., 2011*) for protein. After parameterization, the systems were energy minimized. The systems were then equilibrated in a 10 ns simulation with a NPT ensemble at 310 K temperature and 1 bar pressure. In the equilibration, 1 fs time step was used and harmonic positional restraints of a force constant of 1 kcal/mol/$Å^2$ were initially applied to the protein backbone atoms.

A production simulation was launched using the last frame of the equilibration simulation, with a NPT ensemble at 310 K and 1 bar using the Berendsen coupling scheme (*Berendsen et al., 1984*) with one temperature group and Nosé-Hoover thermostat (*Hoover, 1985*) with a time constant of 0.05 ps. All hydrogen-containing bonds were constrained using a recent implementation (*Lippert et al., 2007*) of the M-SHAKE algorithm (*Krautler et al., 2001*). Reversible reference system propagator algorithm

(r-RESPA) integrator (*Tuckerman and Berne, 1992*) was used. Bonded, Van der Waals, and short-range electrostatic interactions were computed every 2 fs, while long-range electrostatic interactions were computed every 6 fs. All molecular dynamics simulations were performed using Anton, a special purpose supercomputer (*Shaw et al., 2009*) designed for such simulations. The short-range electrostatic interactions were calculated at a cutoff of 9.48 Å, and the long-range electrostatic interactions were computed using Gaussian-split Ewald (*Shan et al., 2005*).

### Structural analysis
Crystal lattice contacts for each crystal form analyzed were identified manually using the program PyMol and written out as kinase pairs. The Lsqkab program in CCP4 was used to superpose the kinase molecules of each pair on each kinase molecule in every other pair to identify any conserved interactions. Buried surface areas for each pair were calculated using Areaimol in CCP4.

## Acknowledgements
We thank Tim Blower, Chris King, and David Meyers for technical assistance and Wei Yang and Scott Bailey for comments on the manuscript. Supported by NIH GM95930 (KH), NIH GM099321 (DJL), NIH GM099092 (DJL and PAC), and Graduate Research Fellowship Program NSF DGE-1232825 (JMM).

## Additional information

### Competing interests
PAC: Reviewing editor, *eLife.* The other authors declare that no competing interests exist.

### Funding

| Funder | Grant reference number | Author |
| --- | --- | --- |
| National Institute of General Medical Sciences | NIH GM099321 | Daniel J Leahy |
| National Institute of General Medical Sciences | NIH GM099092 | Philip A Cole, Daniel J Leahy |
| National Institute of General Medical Sciences | NIH GM95930 | Kalina Hristova |
| Division of Graduate Education | NSF DGE-1232825 | Jacqueline M McCabe |

The funders had no role in study design, data collection and interpretation, or the decision to submit the work for publication.

### Author contributions
JMK, POB, YS, Conception and design, Acquisition of data, Analysis and interpretation of data, Drafting or revising the article; JMMC, Conception and design, Acquisition of data, Analysis and interpretation of data; MKC, ZW, AR, SS, DES, Acquisition of data, Analysis and interpretation of data; KH, PAC, DJL, Conception and design, Analysis and interpretation of data, Drafting or revising the article

### Author ORCIDs
Jacqueline M McCabe, http://orcid.org/0000-0001-9032-5359

## Additional files

### Major datasets
The following previously published datasets were used:

| Author(s) | Year | Dataset title | Dataset ID and/or URL | Database, license, and accessibility information |
| --- | --- | --- | --- | --- |
| Smith BJ, Huang K, Kong G, Chan SJ, Nakagawa S, Menting JG, Hu SQ, Whittaker J, Steiner DF, Katsoyannis PG, Ward CW, Weiss MA, Lawrence MC | 2010 | Structure of the insulin receptor ectodomain, including ct peptide | 3LOH; http://www.rcsb.org/pdb/explore/explore.do?structureId=3loh | publicly available at the RCSB Protein Data Bank. |

| Authors | Year | Title | PDB ID / URL | Availability |
|---|---|---|---|---|
| McKern NM, Lawrence MC, Streltsov VA, Lou MZ, Adams TE, Lovrecz GO, Elleman TC, Richards KM, Bentley JD, Pilling PA, Hoyne PA, Cartledge KA, Pham TM, Lewis JL, Sankovich SE, Stoichevska V, Da Silva E, Robinson CP, Frenkel MJ, Sparrow LG, Fernley RT, Epa VC, Ward CW | 2006 | Insulin receptor (IR) ectodomain in complex with fab's | 2DTG; http://www.rcsb.org/pdb/explore/explore.do?structureId=2dtg | publicly available at the RCSB Protein Data Bank. |
| Menting JG, Whittaker J, Margetts MB, Whittaker LJ, Kong GKW, Smith BJ, Watson CJ, Zakova L, Kletvikova E, Jiracek J, Chan SJ, Steiner DF, Dodson GG, Brzozowski AM, Weiss MA, Ward CW, Lawrence MC | 2013 | Insulin receptor ectodomain construct comprising domains L1,CR,L2, FNIII-1 and alphact peptide in complex with bovine insulin and FAB 83-14 | 3W14; http://www.rcsb.org/pdb/explore/explore.do?structureId=3w14 | publicly available at the RCSB Protein Data Bank. |
| Hubbard SR, Wei L, Ellis L, Hendrickson WA | 1994 | Crystal structure of the tyrosine kinase domain of the human insulin receptor | 1IRK; http://www.rcsb.org/pdb/explore/explore.do?structureId=1irk | publicly available at the RCSB Protein Data Bank. |
| Till JH, Ablooglu AJ, Frankel M, Bishop SM, Kohanski RA, Hubbard SR | 2001 | Crystallographic studies of an activation loop mutant of the insulin receptor tyrosine kinase | 1I44; http://www.rcsb.org/pdb/explore/explore.do?structureId=1i44 | publicly available at the RCSB Protein Data Bank. |
| Favelyukis S, Till JH, Hubbard SR, Miller WT | 2001 | Structure of the insulin-like growth factor 1 receptor kinase | 1K3A; http://www.rcsb.org/pdb/explore/explore.do?structureId=1k3a | publicly available at the RCSB Protein Data Bank. |
| Hubbard SR | 1997 | Phosphorylated insulin receptor tyrosine kinase in complex with peptide substrate and atp analog | 1IR3; http://www.rcsb.org/pdb/explore/explore.do?structureId=1ir3 | publicly available at the RCSB Protein Data Bank. |
| Parang K, Till JH, Ablooglu AJ, Kohanski RA, Hubbard SR, Cole PA | 2001 | Crystal structure of the insulin receptor kinase in complex with a bisubstrate inhibitor | 1GAG; http://www.rcsb.org/pdb/explore/explore.do?structureId=1gag | publicly available at the RCSB Protein Data Bank. |
| Li S, Covino ND, Stein EG, Till JH, Hubbard SR | 2003 | Crystal structure of a catalytic-loop mutant of the insulin receptor tyrosine kinase | 1P14; http://www.rcsb.org/pdb/explore/explore.do?structureId=1p14 | publicly available at the RCSB Protein Data Bank. |
| Velaparthi U, Wittman M, Liu P, Stoffan K, Zimmermann K, Sang X, Carboni J, Li A, Attar R, Gottardis M, Greer A, Chang CY, Jacobsen BL, Sack JS, Sun Y, Langley DR, Balasubramanian B, Vyas D | 2007 | Structure of IGF-1R kinase domain complexed with a benzimidazole inhibitor | 2OJ9; http://www.rcsb.org/pdb/explore/explore.do?structureId=2oj9 | publicly available at the RCSB Protein Data Bank. |
| Munshi S, Hall DL, Kornienko M, Darke PL, Kuo LC | 2003 | Structure of Apo unactivated IGF-1R KInase domain at 1.5A resolution | 1P4O; http://www.rcsb.org/pdb/explore/explore.do?structureId=1p4o | publicly available at the RCSB Protein Data Bank. |
| Munshi S, Kornienko M, Hall DL, Reid JC, Waxman L, Stirdivant SM, Darke PL, Kuo LC | 2002 | Crystal structure of unactivated APO insulin-like growth factor-1 receptor kinase domain | 1M7N; http://www.rcsb.org/pdb/explore/explore.do?structureId=1m7n | publicly available at the RCSB Protein Data Bank. |
| Depetris RS, Hu J, Gimpelevich I, Holt LJ, Daly RJ, Hubbard SR | 2005 | Crystal structure of the Grb14 BPS region in complex with the insulin receptor tyrosine kinase | 2AUH; http://www.rcsb.org/pdb/explore/explore.do?structureId=2auh | publicly available at the RCSB Protein Data Bank. |
| Wu J, Tseng YD, Xu CF, Neubert TA, White MF, Hubbard SR | 2008 | Crystal structure of the insulin receptor kinase in complex with IRS2 KRLB peptide | 3BU3; http://www.rcsb.org/pdb/explore/explore.do?structureId=3bu3 | publicly available at the RCSB Protein Data Bank. |

| Li S, Depetris RS, Barford D, Chernoff J, Hubbard SR | 2005 | Crystal structure of a complex between PTP1B and the insulin receptor tyrosine kinase | 2B4S; http://www.rcsb. org/pdb/explore/explore. do?structureId=2b4s | publicly available at the RCSB Protein Data Bank. |
|---|---|---|---|---|
| Pautsch A, Zoephel A, Ahorn H, Spevak W, Hauptmann R, Nar H | 2001 | IGF-1 receptor kinase domain | 1JQH; http://www.rcsb. org/pdb/explore/explore. do?structureId=1jqh | publicly available at the RCSB Protein Data Bank. |
| Hu J, Liu J, Ghirlando R, Saltiel AR, Hubbard SR | 2003 | Crystal structure of the insulin receptor kinase in complex with the SH2 domain of APS | 1RQQ; http://www.rcsb. org/pdb/explore/explore. do?structureId=1rqq | publicly available at the RCSB Protein Data Bank. |
| Wu J, Li W, Craddock BP, Foreman KW, Mulvihill MJ, Ji QS, Miller WT, Hubbard SR | 2008 | Crystal structure of the insulin-like growth factor-1 receptor kinase in complex with PQIP | 3D94; http://www.rcsb. org/pdb/explore/explore. do?structureId=3d94 | publicly available at the RCSB Protein Data Bank. |
| Mayer SC, Banker AL, Boschelli F, Di L, Johnson M, Kenny CH, Krishnamurthy G, Kutterer K, Moy F, Petusky S, Ravi M, Tkach D, Tsou HR, Xu W | 2008 | Complex structure of insulin-like growth factor receptor and isoquinolinedione inhibitor | 2ZM3; http://www.rcsb. org/pdb/explore/explore. do?structureId=2zm3 | publicly available at the RCSB Protein Data Bank. |
| Patnaik S, Stevens KL, Gerding R, Deanda F, Shotwell JB, Tang J, Hamajima T, Nakamura H, Leesnitzer MA, Hassell AM, Shewchuck LM, Kumar R, Lei H, Chamberlain SD | 2009 | Kinase domain of insulin receptor complexed with a pyrrolo pyridine inhibitor | 3ETA; http://www.rcsb. org/pdb/explore/explore. do?structureId=3eta | publicly available at the RCSB Protein Data Bank. |
| Miller LM, Mayer SC, Berger DM, Boschelli DH, Boschelli F, Di L, Du X, Dutia M, Floyd MB, Johnson M, Kenny CH, Krishnamurthy G, Moy F, Petusky S, Tkach D, Torres N, Wu B, Xu W | 2009 | Complex structure of insulin-like growth factor receptor and 3-cyanoquinoline inhibitor | 3F5P; http://www.rcsb. org/pdb/explore/explore. do?structureId=3f5p | publicly available at the RCSB Protein Data Bank. |

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
