## [Decision Letter]

Thank you for sending your work entitled “Ligand Binding Activates the IGF-1 Receptor by Releasing a Constraint Separating the Transmembrane Regions” for consideration at *eLife.* Your article has been favorably evaluated by John Kuriyan (Senior editor) and 3 reviewers.

The Senior editor and the reviewers discussed their comments before we reached this decision, and the Senior editor has assembled the following comments to help you prepare a revised submission.

This is an excellent paper describing the activation mechanism of the insulin receptor (IR) and insulin-like growth factor receptor (IGF1R), which are homodimeric receptor tyrosine kinases. Because they are constitutively homodimeric, dimerization by itself is excluded as an activation mechanism. Instead, these receptors are thought to be activated by ligand-dependent conformational change, but the nature of the activating transition induced by binding ligand is not yet understood.

Kavran et al. have addressed this problem by analyzing previously published crystal structures and thus identifying a previously unrecognized dimer. The authors propose that this dimer provides important clues as to the conformational changes that take place in the receptor upon insulin binding, and test their hypothesis with experiments designed to probe the effect of ligand binding on the proximity of the dimeric receptor's two transmembrane helices. The authors propose that the transmembrane helices of these receptors are held sufficiently far apart in the absence of ligand to prevent trans-phosphorylation of the kinase domains; ligand binding would stabilize an alternative extracellular domain conformation that allows interaction of the transmembrane segments and, consequently, trans-phosphorylation. The essential message is that ligand binding does not allosterically activate the kinase activity but instead relieves a geometrical constraint that prevents the autophosphorylation that is needed to activate the kinase. The logic is laid out clearly and the experiments in the paper are well presented. Given the importance of this system and the novel model of activation presented, the paper should be of broad interest to *eLife* readers interested in kinase regulation and signal transduction.

Despite the general enthusiasm for the results of this study, the reviewers are concerned about the extent to which the model has been validated (several of the experiments reported in this manuscript do not report directly on this issue). The authors' conclusions rely almost exclusively on FRET between the TM helices, and on disulfide crosslinking experiments intended to show an inherent propensity for TM dimerization. In addition, the paper is quite sparse when it comes to mutagenesis analysis aimed at directly probing the proposed mechanism.

As the authors point out, there are several mechanistic questions about these receptors that remain unanswered: i) what are the constraints on the cytoplasmic domains that prevent trans-phosphorylation in the basal state (TM-helix separation is necessary but is it sufficient)?, ii) how do the juxtamembrane regions contribute to kinase activation?, and iii) what structural communication occurs between the two (equivalent) ligand binding sites when one site becomes occupied? The reviewers recognize that the answers to these questions may be outside the scope of this study.

Focusing just on the issue of transmembrane helix separation, the consensus is that the model needs to be put on firmer footing, and the following suggestions regarding resolving major issues should be considered carefully. It is not our intention to burden the authors with numerous additional experiments, but rather to ask for more evidence that will shift the balance of conviction. The authors are asked to consider the points made below, and to respond by email as to their expectation regarding experiments that are possible to complete in a reasonable timescale, and that would add to the body of evidence. We believe that many of these experiments, if not all, could be done quite quickly. In order to avoid repeated back and forth, it would help to have your response for discussion by the reviewers and to await a response before proceeding with a revision.

Points that are important to address:

1) Mutagenesis of the receptor. To test the hypothesis that the interaction between L1 and ID/Fn3 maintains the basal state, the authors mutated this interface, changing IGF1R residues to IR residues (actual substitutions are not enumerated). This IGF1R mutant was poorly processed/expressed and therefore not amenable to analysis. One questions the rationale for substituting IR residues for IGF1R residues, given that the activation mechanism is probably conserved (was it to insure folding?). As an alternative mutagenesis approach, the authors simply deleted L1. Importantly, the L1-deleted receptor was constitutively active, which may be the first example of an activating mutation (albeit a domain deletion) in the ECD of a (nearly) full-length IGF1R.

The reviewers feel that it is very important that the authors devise and implement a point mutagenesis strategy to disrupt the L1-ID/Fn3 interface and to test the effects of these mutations on activity.

2) Dimerization

The creation of an artificial C905 S-S bond between the IGF-1R beta chains in the extracellular JMD causes beta chain dimerization, as shown. Can the authors show that the formation of this disulfide bond also constitutively activates full length IGF-1R?

The authors could also make additional Cys mutants in the extracellular juxtamembrane region, since this region is likely to be a helical extension of the TMD helix, and moving the Cys one position at a time around the helical wheel would be expected to twist the cytoplasmic domains with respect to one another and determine which S-S dimers have activity. This was done early on for EGFR family members, establishing that only one dimer orientation supports ErbB2 activation (Burke and Stern, MCB 18:5371, 1998).

The authors' model predicts that disulfides near the α-chain C-terminus (αCT) contribute to TM association upon ligand binding. This prediction should be directly testable. As the authors themselves point out, deletion of αCT (as in the construct crystallized by [51], PDB ID: 3W14) should result in: (a) decreased receptor activity; and (b) loss of negative cooperativity in insulin binding. It would be good to see an experiment directly showing these effects.

The authors should test the L1 deletion (IGF1R-delL1-fp) in their FRET assay (Figure 4), which should behave like IGF1R-TM-fp + IGF1.

3) Separating the extracellular domain from the transmembrane domain.

The authors propose that the IGF1R ECD stabilizes the inactive state by maintaining spatial separation of the TMs (and, by extension, separation of the ICDs). This hypothesis is directly testable by expressing full-length receptors with progressively longer flexible linkers between Fn3 and the TM. This experiment could also be performed with linkers on the intracellular side, between the TM and the kinase domain. If the proposed model is accurate, the receptor should become constitutively active when the linker exceeds a certain length. This is not a difficult experiment to perform, especially since the authors were able to confirm constitutive activation of a mutant receptor (L1 deletion).

---

## [Author Response]

*Focusing just on the issue of transmembrane helix separation, the consensus is that the model needs to be put on firmer footing, and the following suggestions regarding resolving major issues should be considered carefully. It is not our intention to burden the authors with numerous additional experiments, but rather to ask for more evidence that will shift the balance of conviction*.

The principal concern expressed by the reviewers/editor was the need for evidence beyond the FRET experiment to “shift the balance of conviction” for the model of Insulin/IGF-1 receptor activation we present, and the major suggestions involved simple experiments to do just that.

We have carried out three new experiments including (i) FRET analysis on the ECD-TM region of the constitutively active ΔL1 IGF1R protein, (ii) cell-based activity assays of IGF1R variants with increasing lengths of flexible linkers between the ECD and TM, and (iii) a combination of cell-based activity and FRET assays on a constitutively active, full-length IGF1R variant with single cysteine (H905C) substituted in its extracellular juxtamembrane region. The results of each experiment conform to predictions based on our model and have been added to our manuscript. We believe that the constitutive activation of the H905C IGF1R variant and its similar FRET efficiency to liganded native IGF1R provides particularly strong evidence that the IGF1R subunit TMs associate in the active state, as well as the surprising result that the active conformation is transiently sampled in the absence of ligand.

Nomenclature: As the nomenclature and specific features of the Insulin/IGF-1 Receptor (IR) family can be daunting, we provide here a brief summary as a reference. IR family members are disulfide-linked homodimers of single-pass integral membrane protein subunits (Figure 10). The extracellular region of each subunit is composed of six subdomains, termed in order L1, CR, L2, Fn1, Fn2, and Fn3, that are followed by transmembrane and intracellular kinase regions. A ∼110 amino acid disordered insertion domain (ID) in the Fn2 domain contains a site of post-translational furin-like cleavage that divides each subunit into α and β chains, which remain linked by a disulfide bond. The αβ subunits, colored orange/blue or white in Figure 10, are linked into αβ homodimers by multiple reciprocal disulfide bonds between α chain cysteines.Author response image 1.(A) IR family schematic and (B) cartoon representation. Disulfide bonds indicated by black diamonds.

Features of IR family members that can complicate mutagenesis experiments include impaired proteolytic processing in variants and difficulties identifying disulfide bonds formed by introduced cysteines owing to the presence of native inter- and intra-subunit disulfides.

*Points that are important to address*:

*1) Mutagenesis of the receptor. To test the hypothesis that the interaction between L1 and ID/Fn3 maintains the basal state, the authors mutated this interface, changing IGF1R residues to IR residues (actual substitutions are not enumerated). This IGF1R mutant was poorly processed/expressed and therefore not amenable to analysis. One questions the rationale for substituting IR residues for IGF1R residues, given that the activation mechanism is probably conserved (was it to insure folding?). As an alternative mutagenesis approach, the authors simply deleted L1. Importantly, the L1-deleted receptor was constitutively active, which may be the first example of an activating mutation (albeit a domain deletion) in the ECD of a (nearly) full-length IGF1R*.

*The reviewers feel that it is very important that the authors devise and implement a point mutagenesis strategy to disrupt the L1-ID/Fn3 interface and to test the effects of these mutations on activity*.

We agree with this point and realize with chagrin that the description of our mutagenesis strategy targeting this region was ambiguous and led to a misunderstanding on the part of the reviewers. To test for the importance of the L1:Fn2’-3’ interface, we noted that we targeted interfacial residues that were conserved between IR and IGF1R for mutagenesis. The reviewers expressed concern that substituting conserved IGF1R residues with IR residues was unlikely to have an effect, which is absolutely correct. What we failed to explain clearly was that although the sites chosen for mutagenesis are conserved (and thus likely important for mediating the interaction and less tolerant of change), the sites were then substituted with non-IR/IGF1R residues that resulted in size/charge reversals, e.g. T to R, E to R, A to R, D to R, G to R, G to E, and L to E, and combinations of up to 8 substitutions were introduced into a single IGF1R variant.

Although a few single substitutions were tolerated, all combinations of substitutions led to loss of proteolytic processing, loss of responsiveness to ligand, and generally lower expression levels. Although anti-phosphotyrosine Western blots showed a higher level of basal phosphorylation for some of these variants, we are hesitant to draw any conclusions from this observation owing to their abnormal expression/processing behavior. One possible explanation for the lowered expression levels is that the mutations led to a constitutively active receptor that is down-regulated by the cell, as is the case for the oncogenic mutations of EGFR (Wang et. al. 2011 NSMB 18:1388). To address this issue we tried to express these IGF1R variants either in the presence of an IGF1R kinase inhibitor or in the context of kinase-inactivating point mutations. Both attempts failed to restore expression or processing. We cannot explain why these variants appear to behave more abnormally than deletion of the entire L1 domain other than to note that proper biogenesis and assembly of IR family members is a complex and evidently delicate process. Receptor formation involves intersubunit dimerization, which is mediated by multiple domains including L1 and Fn2, and proper processing of the ID, which separates synthesis of the Fn2 domain into two halves and must (i) form intersubunit disulfide bonds, (ii) thread around the Fn2 domain, (iii) contribute the αCT helix to the ligand binding site, and (iv) get processed by a furin-like protease. Perhaps owing to these requirements, we commonly observe disrupted biogenesis and processing with seemingly innocuous IGF1R variants.

Although not well described in the initial version of our manuscript, we think our efforts to mutagenize the L1:Fn2’-3’ interface, which include analysis of ten substitutions in various combinations, indicate that analysis of additional variants testing this interface is unlikely to result in a more conclusive outcome. We have revised our manuscript to make the nature and extent of the mutagenesis experiments clearer but hope the editors/reviewers will agree that we should instead focus our efforts on other tests of our model.

2) Dimerization

*The creation of an artificial C905 S-S bond between the IGF-1R beta chains in the extracellular JMD causes beta chain dimerization, as shown. Can the authors show that the formation of this disulfide bond also constitutively activates full length IGF-1R*?

We thank the reviewers for this suggestion. We had thought we had assessed the activity of the H905C IGF1R variant and found it to be normal. When we reviewed our experiments carefully following this suggestion, however, we realized only expression levels, processing, and multimerization of the H905C had been assessed through anti-IGF1R Western blots. Re-examination of this variant using anti-phosphotyrosine Western blots showed that it is clearly constitutively active in the absence of ligand. This result surprised us owing to the large separation between His 905 positions on opposing IGF1R subunits implied by the crystal structure of unliganded Insulin Receptor. We interpret this result to indicate that unliganded IGF1R transiently samples an active conformation in which the TMs are associated (His 905 is just at the extracellular-TM border), and the presence of a cysteine at position 905 traps this active state by forming a reciprocal disulfide bond. An alternative possibility, that inter-IGF1R dimers are formed, is ruled out by Western blots of reducing and non-reducing samples of native and H905C IGF1R showing that the ratio of dimer to monomer is preserved between native and H905C variant IGF1R. That is, we see no appreciable conversion of H905C IGF1R dimers to higher order oligomers. Furthermore, we show constitutively high levels of FRET, comparable to ligand-bound wild type IGF1R between subunits of the H905C variant when its cytoplasmic domains are substituted with a FRET donor-acceptor pair.

We believe these data provide persuasive evidence that (i) IGF1R TMs associate in the active state and (ii) unliganded IGF1R transiently samples the active state. These data are included in the revised version of our manuscript.

*The authors could also make additional Cys mutants in the extracellular juxtamembrane region, since this region is likely to be a helical extension of the TMD helix, and moving the Cys one position at a time around the helical wheel would be expected to twist the cytoplasmic domains with respect to one another and determine which S-S dimers have activity. This was done early on for EGFR family members, establishing that only one dimer orientation supports ErbB2 activation (Burke and Stern, MCB 18:5371, 1998)*.

In addition to the cysteines introduced at positions 905 and 898 in IGF1R TM-icd discussed in the manuscript, we introduced cysteines at positions 903 and 904 but didn’t report on these variants. We observed near complete disulfide bond formation and constitutive phosphorylation in each additional case. The extracellular juxtamembrane regions of IR and IGF1R are not predicted to be alpha helical (PSIPRED), and seven juxtamembrane (JM) proximal residues of IR (homologous to residues 899-905 of IGF1R) are present but not ordered in the crystal structure of the IR ectodomain. Although possible, we are thus not sure this region is helical. I note our results for IGF1R diverge from those results of Burke and Stern on EGFR, who observed very sharp changes in phosphorylation between adjacent sites and strong dimerization for all sites except the two closest to the membrane, comparable to 904 and 905 here.

We are happy to introduce more cysteines at flanking sites and include these data, but I am not sure given the current results that additional experiments would provide conclusive evidence for helicity or influence confidence in our model.

*The authors' model predicts that disulfides near the α-chain C-terminus (αCT) contribute to TM association upon ligand binding. This prediction should be directly testable. As the authors themselves point out, deletion of αCT (as in the construct crystallized by*
[51]*, PDB ID: 3W14) should result in: (a) decreased receptor activity; and (b) loss of negative cooperativity in insulin binding. It would be good to see an experiment directly showing these effects*.

There seems to be a slight misstatement in this comment, which I hope we interpret correctly. The αCT region was not in fact deleted from the crystallized fragment reported by Menting. Fn2, Fn3, and most of the ID region were deleted, but the ∼15 residues encompassing the αCT region were fused to the end of Fn1. What was deleted relevant to this comment were the 3 cysteines in the ID that form intersubunit disulfides that we propose are the likely route by which ligand binding at one site is communicated to the other site to generate negative cooperativity. Our model would predict that loss of the 3 cysteines in the ID would result in loss of negative cooperativity but not loss of activity. Indeed, Surinya et al. (JBC 2002 277:16718) showed that the IR fragment crystallized lacking then Fn2-ID-Fn3 regions bound Insulin with native affinity but did not exhibit negative cooperativity. We have revised our manuscript to make these points explicitly.

*The authors should test the L1 deletion (IGF1R-delL1-fp) in their FRET assay (*Figure 4*), which should behave like IGF1R-TM-fp + IGF1*.

We agree and found that IGF1R ΔL1-fp gives a FRET value (0.38) that is comparable within error to IGF1R ECD-TM-fp in the presence of ligand (0.34) and significantly different than IGF1R ECD-TM-fp in the absence of ligand (0.21) indicating that the conformational changes associated with loss of the L1:Fn2’-3’ interaction are equivalent to the conformational changes following ligand binding. These data have been included in the manuscript.

*3) Separating the extracellular domain from the transmembrane domain*.

*The authors propose that the IGF1R ECD stabilizes the inactive state by maintaining spatial separation of the TMs (and, by extension, separation of the ICDs). This hypothesis is directly testable by expressing full-length receptors with progressively longer flexible linkers between Fn3 and the TM. This experiment could also be performed with linkers on the intracellular side, between the TM and the kinase domain. If the proposed model is accurate, the receptor should become constitutively active when the linker exceeds a certain length. This is not a difficult experiment to perform, especially since the authors were able to confirm constitutive activation of a mutant receptor (L1 deletion)*.

We have now inserted glycine-serine linkers 4, 9, 14, and 20 amino acids in length between the IGF1R extracellular and TM regions and find that linkers of increasing length lead to increasing levels of basal phosphorylation in cell-based activity assays. These data support the notion that ECD enforced TM separation is required for receptor inactivation and have been included in our manuscript.